# A genome-wide screen identifies YAP/WBP2 interplay conferring growth advantage on human epidermal stem cells

Gernot Walko[1,*], Samuel Woodhouse[1,*,†], Angela Oliveira Pisco[1], Emanuel Rognoni[1], Kifayathullah Liakath-Ali[1], Beate M. Lichtenberger[1,†], Ajay Mishra[1,†], Stephanie B. Telerman[1], Priyalakshmi Viswanathan[1], Meike Logtenberg[1,†], Lisa M. Renz[1], Giacomo Donati[1,†], Sven R. Quist[2] & Fiona M. Watt[1]

Individual human epidermal cells differ in their self-renewal ability. To uncover the molecular basis for this heterogeneity, we performed genome-wide pooled RNA interference screens and identified genes conferring a clonal growth advantage on normal and neoplastic (cutaneous squamous cell carcinoma, cSCC) human epidermal cells. The Hippo effector YAP was amongst the top positive growth regulators in both screens. By integrating the Hippo network interactome with our data sets, we identify WW-binding protein 2 (WBP2) as an important co-factor of YAP that enhances YAP/TEAD-mediated gene transcription. YAP and WPB2 are upregulated in actively proliferating cells of mouse and human epidermis and cSCC, and downregulated during terminal differentiation. WBP2 deletion in mouse skin results in reduced proliferation in neonatal and wounded adult epidermis. In reconstituted epidermis YAP/WBP2 activity is controlled by intercellular adhesion rather than canonical Hippo signalling. We propose that defective intercellular adhesion contributes to uncontrolled cSCC growth by preventing inhibition of YAP/WBP2.

[1] Centre for Stem Cells and Regenerative Medicine, Faculty of Life Sciences & Medicine, King's College London, 28th Floor, Tower Wing, Guy's Hospital, London SE1 9RT, UK. [2] Clinic for Dermatology and Venerology, Otto-von-Guericke-University, Leipziger Straße 44, Magdeburg D-39120, Germany. * These authors contributed equally to the work. † Present addresses: Inivata, Li Ka Shing Centre, Robinson Way, CB2 0RE Cambridge, UK (S.W.); Department of Dermatology, Skin and Endothelium Research Division (SERD), Medical University of Vienna, Lazarettgasse 14, 1090 Vienna, Austria (B.M.L.); Cambridge Infinitus Research Centre, Department of Chemical Engineering and Biotechnology, University of Cambridge, Pembroke Street, Cambridge CB2 3RA, UK (A.M.); Division of Immunology, The Netherlands Cancer Institute, Plesmanlaan 121 Postbus 90203, 1006 BE Amsterdam, The Netherlands (M.L.); Department of Life Sciences and Systems Biology, University of Turin, Via Accademia Albertina 13, 10123 Torino, Italy (G.D.). Correspondence and requests for materials should be addressed to F.M.W. (email: fiona.watt@kcl.ac.uk).

Mammalian epidermis comprises a multi-layered epithelium, the inter-follicular epidermis (IFE), which forms the protective interface between the body and the environment, and various epidermal appendages including hair follicles, sebaceous glands and sweat glands[1]. Maintenance of the IFE and its appendages depends on several distinct stem cell (SC) populations[2–4]. IFE SCs reside in the basal cell layer of the epithelium that is anchored to a basement membrane, and divide to produce SCs that remain in the basal cell layer or cells that are destined to undergo terminal differentiation in the suprabasal cell layers (committed progenitor cells (CPs))[1,5].

One of the characteristic tumours of the IFE is cutaneous squamous cell carcinoma (cSCC). These tumours retain some hallmarks of the normal epithelial terminal differentiation programme; however, proliferation is increased, the proportion of differentiated cells is decreased, and the spatial organization of the cell layers is disrupted[6,7]. There is evidence that cSCCs are maintained by a subpopulation of highly proliferative cells termed cancer SCs[8]. These neoplastic SCs appear to hijack the homeostatic controls that operate in normal SCs, eliminating those that promote differentiation and upregulating those that exert a positive effect on proliferation[7].

Primary human epidermal cells and cSCC cells can readily be grown in culture[9,10]. A subset of highy proliferative epidermal cells has the potential to generate large stratified colonies that subsequently fuse to form multi-layered cell sheets, recapitulating the organization of the epidermis[9,11–13]. This culture system has been widely used to study human epidermal SCs and their regulation[11–15], and epidermal sheets generated in vitro are used for autologous transplantation in patients suffering from severe burn wounds or hereditary skin blistering diseases[16,17]. The grafted epidermal sheets can persist as a histologically and physiologically normal epidermis for years[16–18]. However, due to the marked heterogeneity in the proliferative potential of individual primary human epidermal cells[11–13] engraftment of epidermal sheets after transplantation is highly unpredictable[18–20].

In this study, we used an unbiased approach to uncover the molecular basis for this heterogeneity by performing genome-wide pooled RNA interference (RNAi) screens in normal epidermal cells and neoplastic (cSCC) cells with increased growth potential. This led us to identify the Hippo effector YAP and its co-factor WBP2 as drivers of clonal expansion of normal and neoplastic human epidermal SCs via TEAD transcription factors. By examining the functions of YAP and WBP2 and their upstream regulators we provide new evidence for the role of canonical and non-canonical Hippo signalling in normal and neoplastic epidermis.

## Results

**Genome-wide RNAi screen.** To identify genes that confer a clonal growth advantage on primary human epidermal cells (normal human keratinocytes; NHKs), we conducted a genome-wide RNAi screen, using pooled short-hairpin RNAs (shRNAs) (Fig. 1a,b). We screened 82,305 shRNAs targeting 15,256 protein-coding genes (Fig. 1a,b and Supplementary Data 1), with each gene being targeted by 5–6 individual shRNAs. The abundance of each genome-integrated barcoded lentiviral hairpin was quantified from genomic DNA by Illumina deep sequencing[21] (Fig. 1c) in the initial cell population (24 h after lentiviral transduction; $t = 0$) and after 14 days of culture at a clonal seeding density on a 3T3-J2 fibroblast feeder layer ($t = 14$), by which time the cells had formed a confluent, stratified sheet (Fig. 1a,b). Under these culture conditions individual epidermal SCs initially give rise to exponentially growing (expanding) colonies[11] that eventually become multi-layered as differentiating cells in the colony

stratify[9,11,13], causing a switch in the growth behaviour in the colony centre to a 'balanced' mode where similar proportions of proliferating and differentiating cells are generated[11] (Supplementary Movie 1). In contrast, CP cells (transit amplifying cells) form abortive colonies with a very limited growth capacity due to early onset of terminal differentiation[12,13] (Supplementary Movie 2). Since the bulk growth of a NHK population in culture is driven by the SC fraction[11–13], the majority of hits identified in the screen should represent genes that promote clonal expansion of SCs (Fig. 1b) rather than genes that directly regulate the onset of terminal differentiation. We focused on shRNAs that were significantly under-represented at $t = 14$, since they should target genes that positively regulate SC proliferation and survival or negatively regulate the transition from SC to CP cell (Fig. 1b).

To complement the NHK screen we introduced the shRNA library into SCC13 cells, a cell line established from a cSCC[10] that fails to undergo terminal differentiation in culture[22]. Compared with NHKs, SCC13 cells have a much higher colony forming efficiency[22], consistent with an expansion of their SC pool. We measured the abundance of each genome-integrated barcoded lentiviral hairpin in SCC13 cells at $t = 0$ and after 14 days in culture after plating at clonal density, by which time the cells formed a confluent, partially multi-layered cell sheet (Fig. 1a–c).

Gene target hits from both screens were identified using a bioinformatics pipeline (see Supplementary Methods, Supplementary Figs 1,2,3a,b, Supplementary Data 2). We identified 641 and 6,109 candidate genes essential for NHK and SCC13 clonal growth, respectively (Fig. 1d,e, Supplementary Fig. 3c,d and Supplementary Data 3). Out of these, 396 and 5,836 represented candidate positive NHK and SCC13 growth regulators, respectively, and were significantly enriched for Gene Ontology (GO) terms and gene sets associated with normal cell growth (Supplementary Fig. 3e and Supplementary Data 4,5). Genes encoding proteins involved in housekeeping functions, including 60S and 40S ribosomal proteins, were highly represented among the hits for both NHKs and SCC13 cells (Supplementary Fig. 3f and Supplementary Data 3–5). Indeed, the top 10 gene target hits included multiple ribosomal genes and genes essential for messenger RNA splicing and protein synthesis (Fig. 1f,g). As expected, we did not detect any factors known to directly induce terminal differentiation, such as AP1 transcription factors[23], among the candidate negative regulators of NHK growth (Supplementary Data 6). We did however detect known suppressors of NHK proliferation, including *E2F5* (ref. 24) and *POU3F1*, a transcriptional repressor of *TP63* (ref. 25 and Supplementary Data 4).

The SCC13-specific candidate growth regulators were significantly enriched for GO terms associated with cancer (Supplementary Fig. 3g,h). AP1 transcription factors and WNT signalling components, both known to be important for cSCC growth in vivo[21,23,26], were highly represented, as was a cSCC gene signature compiled from clinically annotated and transcriptionally profiled human samples (Supplementary Fig. 3i and Supplementary Data 6,7). These findings are consistent with the deregulation of multiple signalling pathways and transcription factors in SCC[27] and confirm the power of the screen to identify regulators of neoplastic growth/viability.

**Identification of YAP as a positive growth regulator.** The top candidate for positive regulation of NHK growth was *YAP1* (Fig. 1f and Supplementary Data 3), encoding the Hippo signalling effector and transcriptional co-activator YAP[28,29]. YAP was also present among the 328 candidate growth regulators common to both NHKs and SCC13 cells (Supplementary Data 3).

The YAP paralogue TAZ (*WWTR1*) was a candidate SCC13 cell-specific growth regulator (Supplementary Data 3), consistent with the observation that TAZ is considerably more abundant in cSCC cells than NHKs (Supplementary Fig. 4a). Despite compensatory upregulation of TAZ, YAP knockdown was more effective than

TAZ knockdown in reducing clonal growth of SCC13 cells (Supplementary Fig. 4b,c), and YAP was highly expressed in putative cSCC-associated SCs (Supplementary Fig. 4d,e). YAP acts predominantly through transcription factors of the TEAD (TEA domain) family[30,31]. Consistent with this we detected

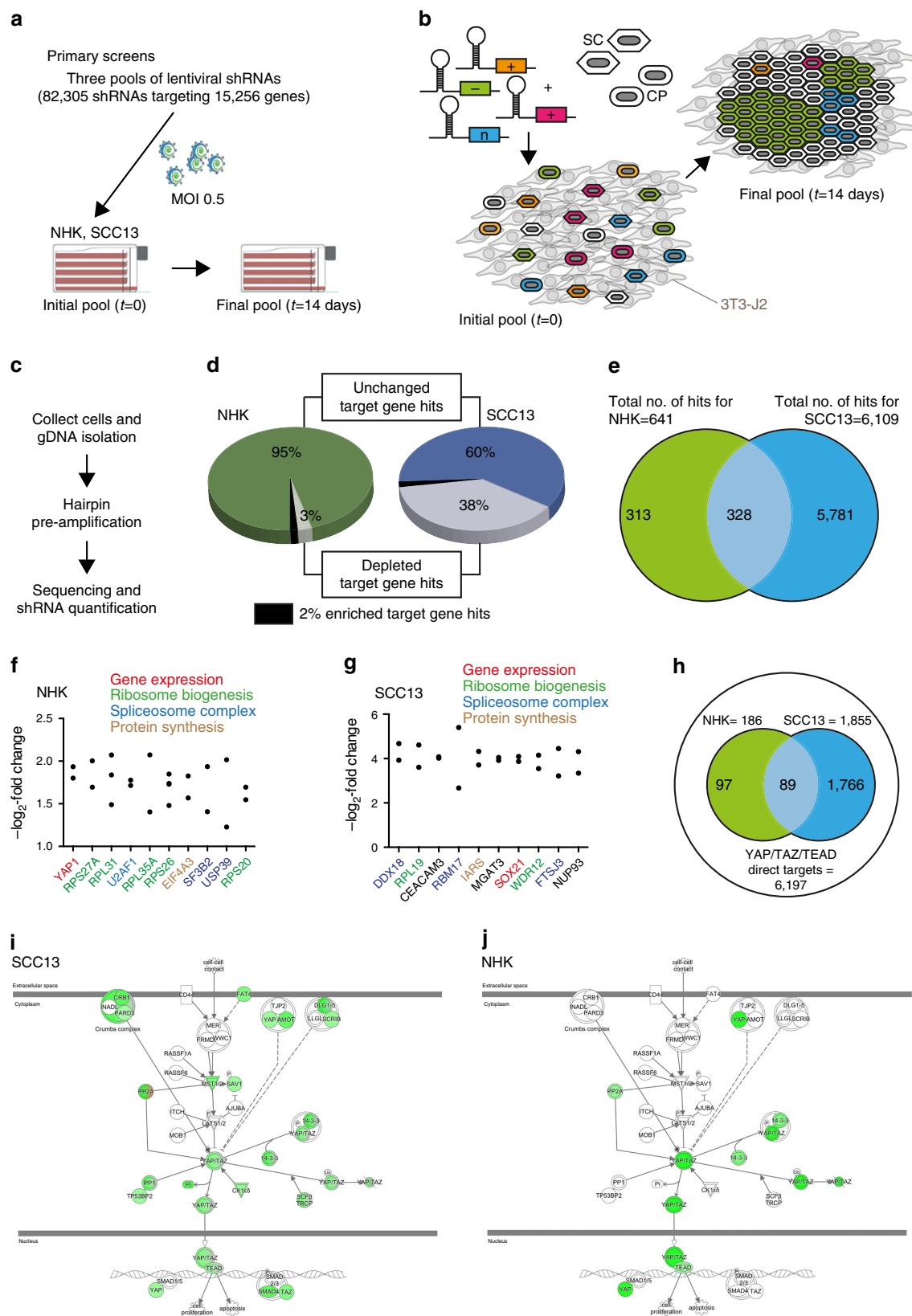

*TEAD4* among the candidate positive NHK growth regulators and *TEAD3* and *TEAD4* among the candidate positive SCC13 cell growth regulators (Supplementary Data 3). Further underscoring the importance of YAP/TEAD, we found multiple previously identified YAP/TAZ/TEAD target genes[30,31] among the candidate positive growth regulators in our screen and in the cSCC gene signature (Fig. 1h, Supplementary Fig. 3i and Supplementary Data 7,8). Thus, our screens identified YAP/TAZ/TEAD as a key regulator of clonal growth of human epidermal- and neoplastic (cSCC) SCs *in vitro*.

To explore the mechanistic networks controlling YAP activity, we first curated the Hippo network interactome by combining published proteomic data sets[32–34] with the list of Hippo pathway components obtained from Ingenuity. We filtered these 664 Hippo network components for genes that were identified as candidate regulators of clonal growth of NHK and SCC13 cells in the genome-wide shRNA screens (Supplementary Data 9). This identified multiple (canonical and non-canonical) Hippo network components as candidate regulators of clonal SCC13 growth (Fig. 1i and Supplementary Data 9), but did not provide any evidence of involvement of canonical Hippo signalling in regulating YAP activity to promote clonal NHK growth (Fig. 1j and Supplementary Data 9).

To evaluate canonical Hippo signalling experimentally, we examined expression of LATS1, one of the two core canonical Hippo pathway kinases acting immediately upstream of YAP/TAZ[28,29], which phosphorylates YAP on S127. LATS1 was strongly expressed in SCC13 cells but not in NHKs under growth-promoting culture conditions (Supplementary Fig. 4f). Consistent with a role of canonical Hippo signalling in suppressing YAP upon terminal differentiation[35], we observed that when terminally differentiated cells accumulated in stratifying cultures, the levels of total and activated LATS1 (auto-phosphorylated at S909 (ref. 28)) and pS127 YAP increased in NHKs, and there was a concomitant downregulation of total YAP (Supplementary Fig. 4f). This suggested that canonical Hippo signalling becomes activated during NHK terminal differentiation to downregulate YAP.

YAP/TAZ are known to be regulated in a non-canonical manner by WNT signalling[36,37]. Consistent with WNT signalling pathway components not being represented among the candidate positive NHK growth regulators, we found that treatment of NHKs with the WNT antagonist IWP2 had no effect on YAP localization and clonogenic growth potential (Supplementary Fig. 4g,h). Thus our screens provided no evidence for an involvement of WNT signalling in regulating nuclear localization of YAP.

**WBP2 interacts with YAP to regulate NHK growth**. To examine factors that modulate YAP/TEAD activity in NHKs we chose 10 candidate genes identified from the genome-wide screen that represent non-canonical Hippo network components (Fig. 2a and Supplementary Data 10). Four of these genes (*AMOTL2*, *ENY2*, *DVL1* and *NYNRIN*) were identified as negative growth regulators in the screen and thus represented candidate YAP suppressors, while the others represented candidate YAP activators (Fig. 2a). We silenced the expression of each of the 10 candidate genes in NHKs using a customized SMARTpool siRNA library (Fig. 2b,c). siRNAs targeting *YAP1* and *TAZ* were used as positive and negative controls, respectively, in addition to non-targeting control siRNAs. As expected, knockdown of YAP, but not TAZ, reduced TEAD-mediated transcription, as measured by reduced transcript levels of the bona fide YAP/TAZ/TEAD target genes *CTGF*, *CYR61* and *BIRC5* (refs 30,31,38, Fig. 2c and Supplementary Fig. 5a). Although LATS1 was upregulated during NHK differentiation (Supplementary Fig. 4f), knockdown of LATS1 or 2, individually or in combination, did not affect mRNA levels of *CTGF* and *CYR61* (Fig. 2c, Supplementary Fig. 5a,f,g). Only one siRNA SMARTpool, targeting WW-binding protein 2 (*WBP2*), had a robust effect on YAP transcriptional co-activator functions (Fig. 2c and Supplementary Fig. 5a). Deconvolution of the WBP2 and YAP siRNA SMARTpools revealed that each individual siRNA in the pools was capable of significantly downregulating expression of the respective mRNA and of causing a concomitant decrease in CYR61 expression (Supplementary Fig. 5b–d). Similar results were obtained using shRNAs (Supplementary Fig. 5e–g).

WBP2 was shown in *Drosophila* to cooperate with the YAP homologue Yorkie to drive tissue growth[39], and in human breast epithelial cells to promote the transforming ability of TAZ[40]. However, its functions in mammalian epidermis are unknown. Using *in situ* proximity ligation assays (PLA), we confirmed an interaction of YAP with WBP2 in the nucleus and cytoplasm of NHKs (Fig. 2d,e and Supplementary Fig. 5i,j). Nuclear abundance of the YAP/WBP2 complex was higher in SCC13 cells than NHKs (Fig. 2f). YAP, but not TEADs, could be co-immunoprecipated with WBP2 (Fig. 2g). However, in the nucleus, both YAP and WBP2 were in close proximity to TEAD1 (Fig. 2h,i). Knockdown of WBP2 impaired transcriptional activation of a synthetic TEAD reporter (8xGTIIC-Lux) to a comparable extent to YAP knockdown (Fig. 2j), while not affecting the interaction between YAP and TEADs in the nucleus of NHKs (Supplementary Fig. 5k–n). Likewise, overexpression of WBP2 in NHKs increased CYR61 and CTGF expression in a YAP-dependent manner, but did not affect the expression levels of YAP (Fig. 2k,l).

**Figure 1 | Genome-wide screens for growth regulators of normal keratinocytes and SCC13 cells.** (**a**,**b**) Schematic representation of the RNAi screens based on relative enrichment/depletion of individual shRNAs over time. Primary human keratinocytes (NHKs, comprising SCs, CP cells, and terminally differentiated cells) or SCC13 cells are transduced with the shRNA library, and shRNA abundance is analysed 24 h after infection ($t = 0$). Genes that regulate clonal growth are then identified in separate screens, in which infected cells are seeded at clonal density into multilayer flasks containing a fibroblast feeder layer (3T3-J2) and allowed to grow into colonies over 14 days ($t = 14$). shRNAs targeting essential positive ($+$) growth regulators are expected to become under-represented at $t = 14$ due to reduced SC growth/survival, while shRNAs targeting negative ($-$) growth regulators will become over-represented at $t = 14$. shRNAs targeting non-essential ($n$) growth regulators will not change in abundance. (**c**) After isolation of genomic DNA (gDNA), individual shRNAs are pre-amplified and quantified by sequencing unique 18 bp barcode sequences. (**d**) Pie chart summaries of the shRNA-targeted gene hits identified with our bioinformatics pipeline. Unchanged category: all gene targets whose respective shRNA abundances were not significantly changed ($P > 0.05$) at $t = 14$ or for which the average $\log_2$-fold change in abundance of respective significant shRNAs was lower than the cutoff of 0.7 or for which the pattern of change in abundances of respective significant shRNAs was ambiguous. Enriched and depleted: gene targets whose respective shRNAs where significantly enriched or depleted, respectively, at $t = 14$. (**e**) Venn diagram depicting total numbers of unique and shared gene target hits. (**f**,**g**) shRNAs for top ten essential regulators of clonal NHK (**f**) and SCC13 (**g**) growth are markedly depleted at $t = 14$. (**h**) Venn diagram depicting the numbers of gene target hits representing YAP/TAZ/TEAD direct target genes. (**i**,**j**) Representation of known Hippo network components among genes identified as candidate regulators of SCC13 (**i**) and NHK (**j**) growth. Colour scheme identifies genes which were identified as candidate positive (green) or negative (red) growth regulators. Colour intensities reflect the average $\log_2$-fold change in shRNA abundance of the respective gene targets. Nodes coloured green and grey or red and grey indicate that one or more components were not present in our data set.

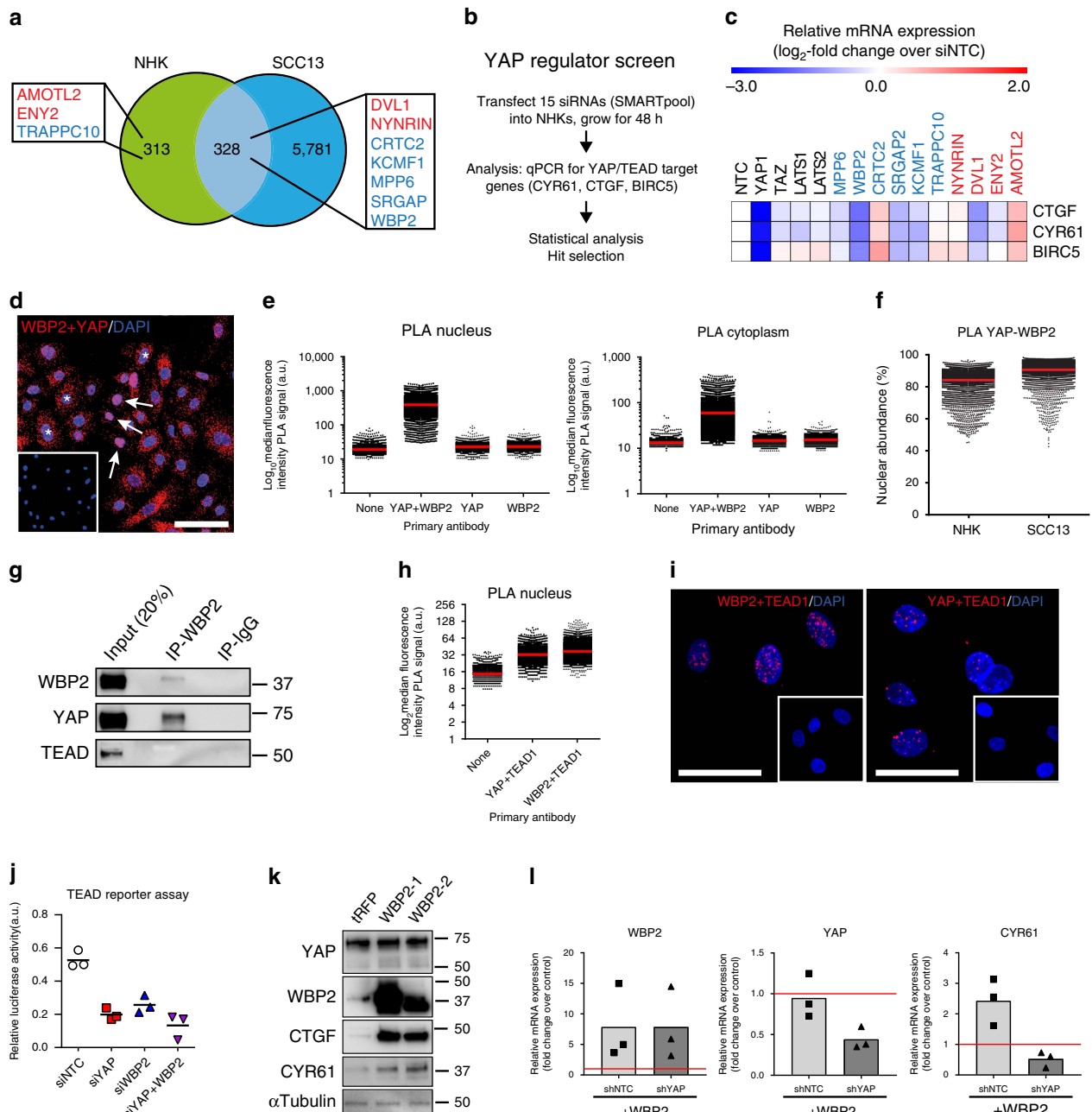

**Figure 2 | Identification of WBP2 as a YAP co-factor enhancing YAP/TEAD-mediated gene transcription.** (**a**) Candidate non-canonical YAP regulators (identified in the screens). Blue: putative YAP activators (shRNAs under-represented at $t = 14$), red: putative YAP suppressors (shRNAs over-represented at $t = 14$). (**b**) Screening strategy. (**c**) Heat map shows mRNA expression levels of YAP/TAZ/TEAD target genes in NHKs transfected with siRNA SMARTpools targeting YAP regulators identified in (**a**) or non-targeting (NTC) siRNA. qRT-PCR data are from five independent experiments (normalized to 18sRNA). (**d**) *In situ* PLA for endogenous YAP/WBP2 interactions in NHKs. Cells with predominantly nuclear (arrows) or cytoplasmic (asterisks) YAP/WBP2 interactions are shown. Insert, control (primary antibodies omitted). Scale bar, 50 μm. (**e,f**) Quantification of YAP/WBP2 PLA signals (individual data points) in NHKs (**e,f**) or SCC13 cells (**f**). One representative experiment with >10,000 cells per condition is shown. Red lines represent mean. (**g**) Western blotting of immunoprecipitates and input lysates with antibodies shown. (**h**) Quantification of PLA signals (individual data points) in NHKs. One representative experiment with 8,000 cells per condition is shown. Red lines represent mean. (**i**) Representative images (maximum intensity projections) of *in situ* PLA detection of endogenous YAP/TEAD1 and WBP2/TEAD1 interactions in NHKs. Inserts, controls (no primary antibodies). Scale bars, 50 μm. (**j**) Luciferase reporter assay (8xGTIIC) in HEK-293 T cells transfected with the indicated siRNA SMARTpools. Data are from three biological replicates (independent siRNA tranfections, individual data points and their mean) from one representative experiment. (**k**) Western blot analysis of NHKs stably over-expressing two different open reading frames of WBP2 (WBP2-1, WBP2-2) or TurboRFP (tRFP) as control, using antibodies shown. Tubulin was used as loading control. (**l**) qRT-PCR analysis of NHKs co-expressing either a YAP-specific shRNA or a non-targeting control shRNA (shNTC) together with WBP2 (open reading frame WBP2-1). NHKs expressing shNTC but not over-expressing WBP2 served as control. Data are from three independent experiments performed with two biological replicates (independent lentiviral infections). Individual data points represent average fold change in mRNA abundance (normalized to 18sRNA) compared with control (red line) in each experiment. Bars represent the mean. qRT-PCR, quantitative real-time PCR.

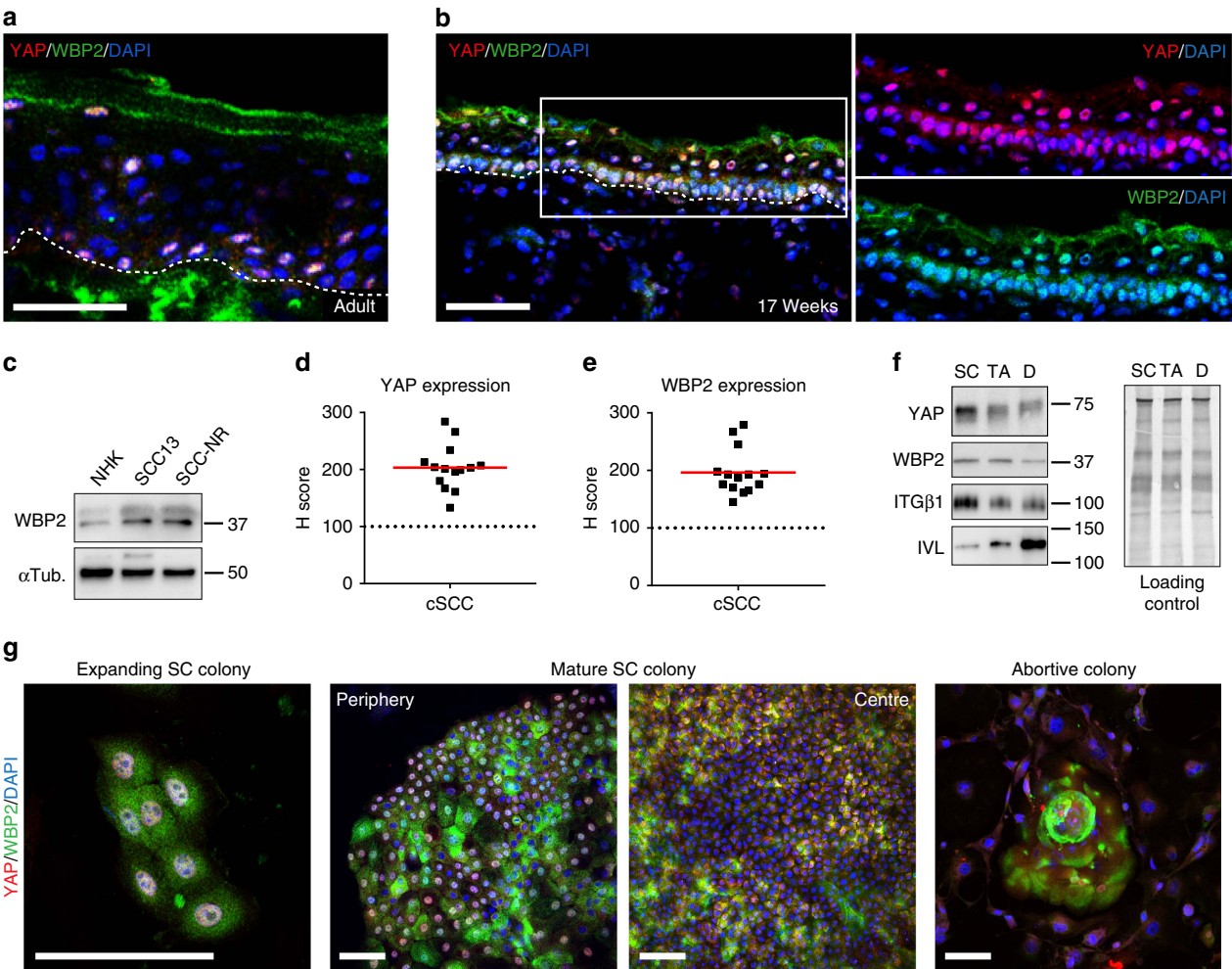

**Figure 3 | YAP and WBP2 are highly expressed in human epidermal SCs and upregulated in cSCC.** (**a,b**) Representative images (single optical planes) of adult (**a**) and fetal (17 weeks gestation) (**b**) human skin sections immunolabelled with the indicated antibodies and counterstained with DAPI to reveal nuclei. White dotted lines demarcate dermal-epidermal boundaries. Scale bars, 50 μm. (**c**) Western blot analysis of NHKs, SCC13 cells and primary cSCC cells (SCC-NR) using antibodies against WBP2. Tubulin was used as loading control. (**d,e**) Semiquantitative analysis (H-score) of YAP (**d**) and WBP2 (**e**) immunostaining intensities (individual data points) in a panel of cSCC- compared with normal skin sections (dotted line). Red lines represent the mean. (**f**) Western blot analysis of enriched populations of stem- (SC), transit amplifying- (TA), and terminally differentiated (D) cells using antibodies against YAP, WBP2, ITGβ1, and involucrin (IVL). Equal protein loading was confirmed by enhanced tryptophan fluorescence imaging (Bio-Rad) of PVDF membranes (loading control). (**g**) Representative images (maximum intensity projections) of expanding and mature SC colonies as well as abortive colonies, immunolabelled with indicated antibodies and counterstained for DAPI to reveal nuclei. Scale bars, 100 μm. PVDF, polyvinylidene difluoride.

We conclude that WBP2 acts as a co-factor of YAP, enhancing TEAD-mediated transcription without modulating YAP-TEAD interaction.

**Expression of YAP/WBP2 in NHKs and cSCC cells**. It is well documented that YAP and TAZ are required for normal epidermal development and hair growth in the mouse[30,41,42], and YAP and TAZ drive pro-tumorigenic signals in mouse and human cSCC[30,43]. Nevertheless, there is ambiguity about YAP/TAZ function: YAP and TAZ are largely dispensable for homoeostasis of the IFE in adult mouse skin[30,42], and in cSCC both growth-promoting and growth-inhibiting functions have been documented[43,44]. This led us to hypothesize that nuclear YAP/WBP2 abundance (and thus YAP/WBP2/TEAD-mediated transcriptional activity) might differ in different subpopulations of NHKs, according to their proliferative potential.

We observed heterogeneous nuclear abundance of both YAP and WBP2 in basal cells in adult human epidermis (Fig. 3a). This contrasted with fetal skin, where YAP and WBP2 were predominantly nuclear in the majority of basal epidermal cells (Fig. 3b), likely reflecting the higher proportion of proliferative keratinocytes in fetal epidermis[45]. Similarly, nuclear YAP and WBP2 were prominent in the basal layer of embryonic and neonatal mouse IFE (Supplementary Fig. 6a, E17.5 and P0), but were less abundant in adult IFE (Supplementary Fig. 6a, P21). During wound healing of adult mouse skin there was a dramatic increase in nuclear YAP and WBP2 within the epidermis (Supplementary Fig. 6b). We conclude that abundant nuclear YAP/WBP2 in human and mouse IFE correlates with a high level of proliferation.

Consistent with the higher clonogenic growth potential of cSCC cells compared to NHKs, we found the protein levels of YAP and WBP2 to be increased in SCC13 and primary cSCC cells compared with NHKs (Fig. 3c and Supplementary Fig. 4a), and we also observed increased expression of YAP and WBP2 in a panel of primary human cSCCs (Fig. 3d,e and Supplementary Fig. 7). When we fractionated NHK cultures by differential adhesion to extracellular matrix[12], expression of YAP and WBP2

was highest in the rapidly adhering, SC-enriched, fraction displaying the highest colony formation efficiency (CFE) (Fig. 3f and Supplementary Fig. 8a). Underscoring these findings, we also detected YAP and WBP2 among a set of genes that are actively transcribed from human epidermal SC-specific enhancers (Supplementary Data 11 and 12)[46,47].

We observed nuclear YAP/WBP2 in all cells of expanding NHK colonies that were founded by SCs, but mostly cytoplasmic YAP/WBP2 in abortive colonies comprising terminal differentiated cells (Fig. 3g and Supplementary Fig. 8b). In large, mature SC colonies, YAP and WBP2 were found to be nuclear in most cells at the still actively expanding periphery, but cytoplasmic in the majority of cells in the colony centre consistent with the switch to a 'balanced' growth mode in response to local contact-inhibition[11] (Fig. 3g and Supplementary Fig. 8b). By constructing a micro-epidermis on 80 μm diameter micro-patterned circular

adhesive islands[48], we could observe that YAP was specifically downregulated in terminally differentiating cells (Supplementary Fig. 8c). Taken together these data suggest that YAP and WBP2 are highly expressed in human epidermal SCs and upregulated in cSCC, and that nuclear YAP/WBP2 localization indicates an actively proliferating SC state.

**Loss of WBP2 results in reduced epidermal proliferation.** To examine the function of WBP2 *in vivo*, we examined the skin of WBP2 knockout mice[49]. All of the differentiated layers of the IFE formed normally and there were no obvious hair follicle abnormalities (Supplementary Fig. 9a,b). Consistent with YAP/TAZ being upregulated during epidermal development and wound healing, rather than in steady-state adult epidermis[30,41,42], we observed reduced IFE proliferation in neonatal (P5) but not adult WBP2 knockout mice (Fig. 4a,b and Supplementary

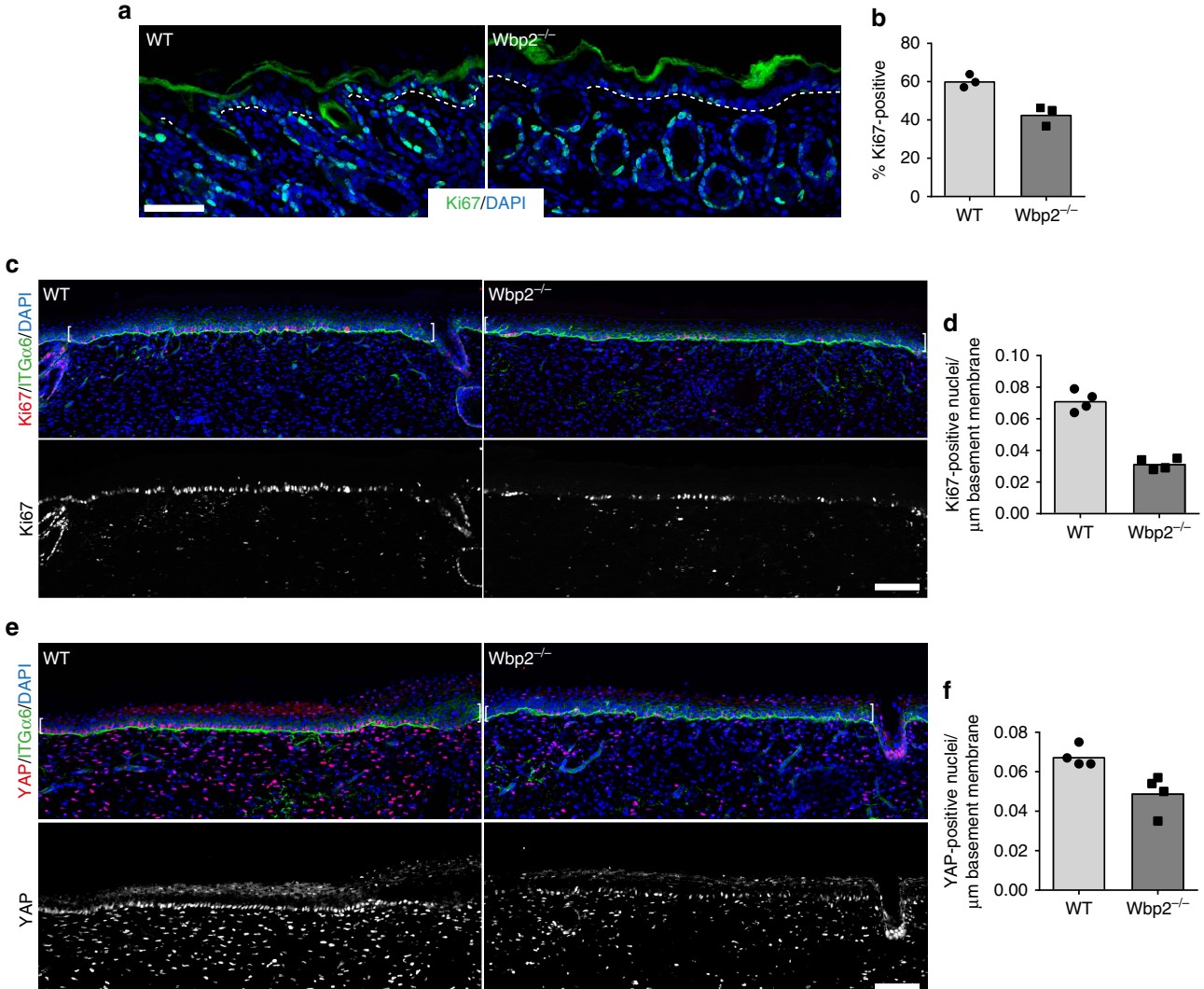

**Figure 4 | WBP2 promotes epidermal regeneration *in vivo*. (a)** Representative images of wild-type (WT) and WBP2-null (Wbp2$^{-/-}$) mouse back skin samples at postnatal day 5. Sections were immunolabelled with the indicated antibodies and counterstained with DAPI to reveal nuclei. White dotted lines demarcate dermal-epidermal boundaries of the IFE. Scale bars, 50 μm. **(b)** Proliferative indices of IFE at postnatal day 5. Individual data points represent the average percentage of Ki67-positive basal epidermal cells, quantified from two immunolabelled sections representing different back skin areas. Bars represent the mean. **(c,e)** Representative images of wild-type (WT) and WBP2-null (Wbp2$^{-/-}$) mouse back skin samples at 14 days after full-thickness wounding. Sections were immunolabelled with the indicated antibodies and counterstained with DAPI to reveal nuclei. Brackets denote the wound bed area. Scale bars, 100 μm. **(d,f)** Quantification of basal cells positive for Ki67 **(d)** or nuclear YAP **(f)** in regenerating epidermis. Individual data points represent the average number of Ki67- or nuclear YAP-positive basal epidermal cells per μm of basement membrane, quantified from four immunolabelled sections representing different parts of the wound bed. Bars represent the mean.

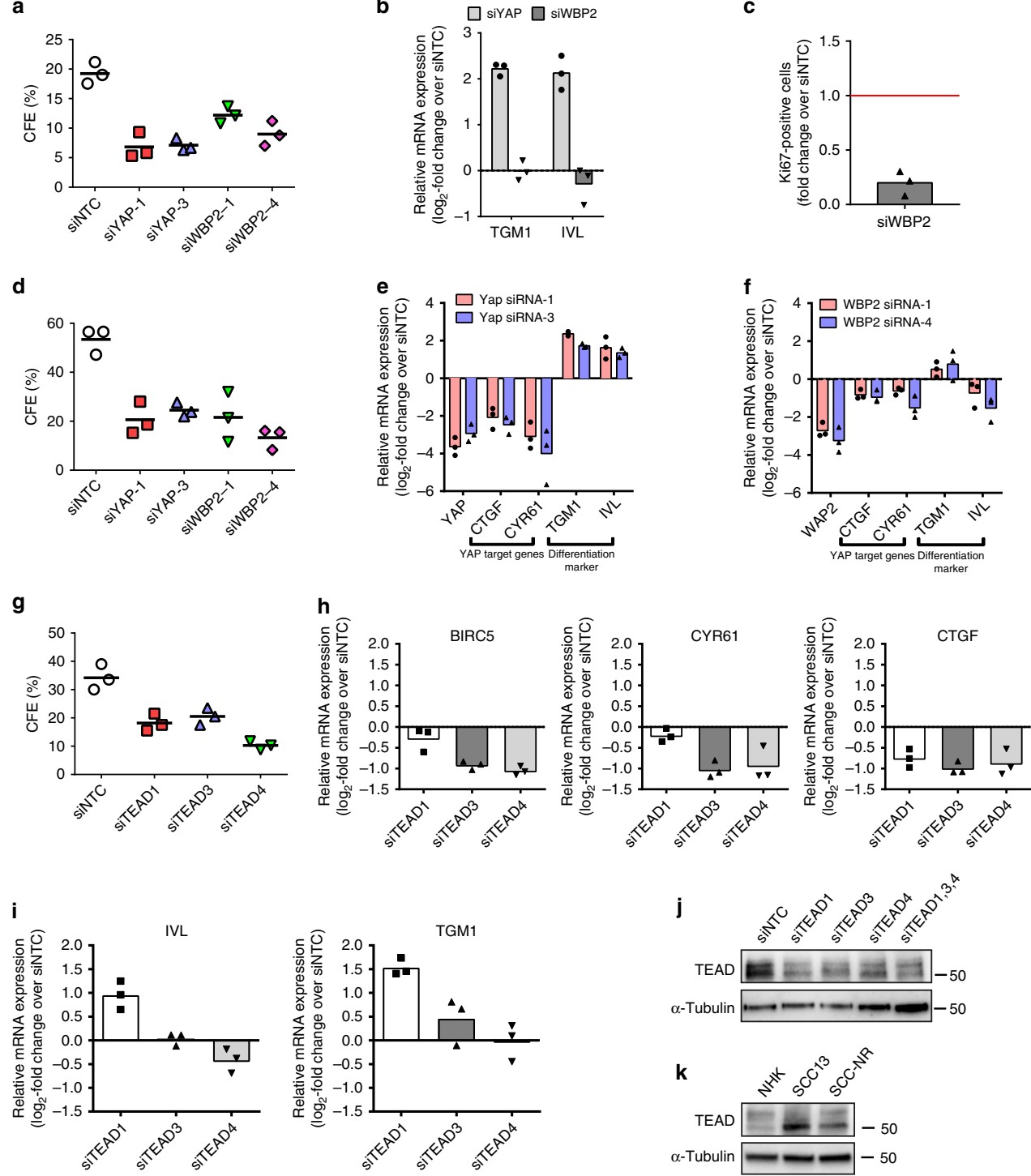

**Figure 5 | YAP, WBP2, and TEADs drive normal- and neoplastic epidermal SC proliferation.** (**a**,**d**,**g**) Clonal growth assays of NHKs (**a**,**g**) and SCC13 cells (**d**) transfected with the siRNAs shown. Data shown are from three independent experiments performed with three technical replicates. Individual data points represent average percentage of colonies formed per number of cells seeded (CFE); lines represent the mean. (**b**,**e**,**f**,**h**,**i**) qRT-PCR analysis of YAP/TAZ/TEAD target genes and terminal differentiation marker expression in NHKs (**b**,**h**,**i**) and SCC13 cells (**e**,**f**) transfected with the specific siRNA SMARTpools indicated or a non-targeting control (NTC) siRNA. Data shown are from three independent experiments performed with two biological replicates (independent siRNA transfections). Individual data points represent the average log$_2$-fold change in mRNA abundance (normalized to 18sRNA) compared with siNTC in each experiment. Bars represent the mean. (**c**) Quantification of cell proliferation in NHKs transfected with WBP2-specific siRNA SMARTpools or a non-targeting control (NTC) siRNA. Data shown are from three independent experiments. Individual data points represent the fold change in Ki67-positive cells compared with control (siNTC, red line) in each experiment, bar represents the mean. More than 500 cells were analysed per experiment. (**j**) Western blot analysis of NHKs transfected with siRNAs silencing individual TEAD proteins, using antibodies against pan-TEAD. Tubulin was used as loading control. (**k**) Western blot analysis of NHKs, SCC13 cells and primary cSCC cells (SCC-NR) growing as clonal colonies on a fibroblast feeder layer (which was removed before cell lysis) using antibodies against pan-TEAD. Tubulin was used as loading control. qRT-PCR, quantitative real-time PCR.

Fig. 9c). There was a significant reduction in proliferating basal cells during IFE regeneration of WBP2 knockout compared with wild-type mice (Fig. 4c,d). This correlated with reduced numbers of basal cells with nuclear YAP (Fig. 4e,f). These results reveal an important role of WBP2 in promoting YAP/TEAD-mediated epidermal proliferation during development and wound healing.

**Role of YAP/WBP2 in regulating NHK and cSCC cell growth.** Knockdown of YAP or WBP2 reduced the clonal growth of NHKs (Fig. 5a). In contrast to YAP knockdown, WBP2 knockdown was not associated with induction of terminal differentiation (Fig. 5b), but solely with a decrease in proliferation (Fig. 5c). Knockdown of WBP2, like YAP, also impaired the clonal growth of SCC13 cells (Fig. 5d). Whereas knockdown of YAP stimulated terminal differentiation in SCC13 cells, as indicated by de-novo expression of *TGM1* and *IVL*, knockdown of WBP2 did not (Fig. 5e,f).

Consistent with YAP/WBP2 acting through TEAD transcription factors[30,31], knockdown of TEAD proteins expressed in NHKs also impaired clonal growth (Fig. 5g). Knockdown of TEAD3 and TEAD4 reduced expression of the YAP/TAZ/TEAD target genes *CTGF*, *CYR61* and *BIRC5* to a similar extent and

did not significantly impact on the expression of terminal differentiation markers. In contrast, knockdown of TEAD1 induced terminal differentiation but only reduced expression of CTGF (Fig. 5h–j). TEAD proteins were more abundant in SCC13 cells compared with NHKs (Fig. 5k). Together our studies establish YAP, WBP2 and TEADs as key regulators of proliferation of human epidermal SCs *in vitro* and suggest differential roles in the regulation of terminal differentiation.

The physiological relevance of YAP/WBP2 in human IFE was verified by epidermal reconstitution assays in which cells were seeded onto de-epidermised dermis and cultured at the air–liquid interface[15]. After 3 weeks control cells had reconstituted a morphologically normal IFE with distinct basal, spinous, granular and cornified layers. In contrast, NHKs expressing YAP or WBP2-specific shRNAs were only capable of reconstituting a hypoplastic epidermis with reduced proliferation (Fig. 6a–c and Supplementary Fig. 5e–g). Consistent with our *in vitro* data, there was premature onset of terminal differentiation in the basal cell compartment in the case of YAP knockdown epidermis (Fig. 6a). In contrast, knockdown of LATS1/2 had no effect on epidermal reconstitution (Fig. 6a–c and Supplementary Fig. 5e–g).

Our data suggest that basal epidermal cells in which YAP has been silenced will stop proliferating and undergo terminal

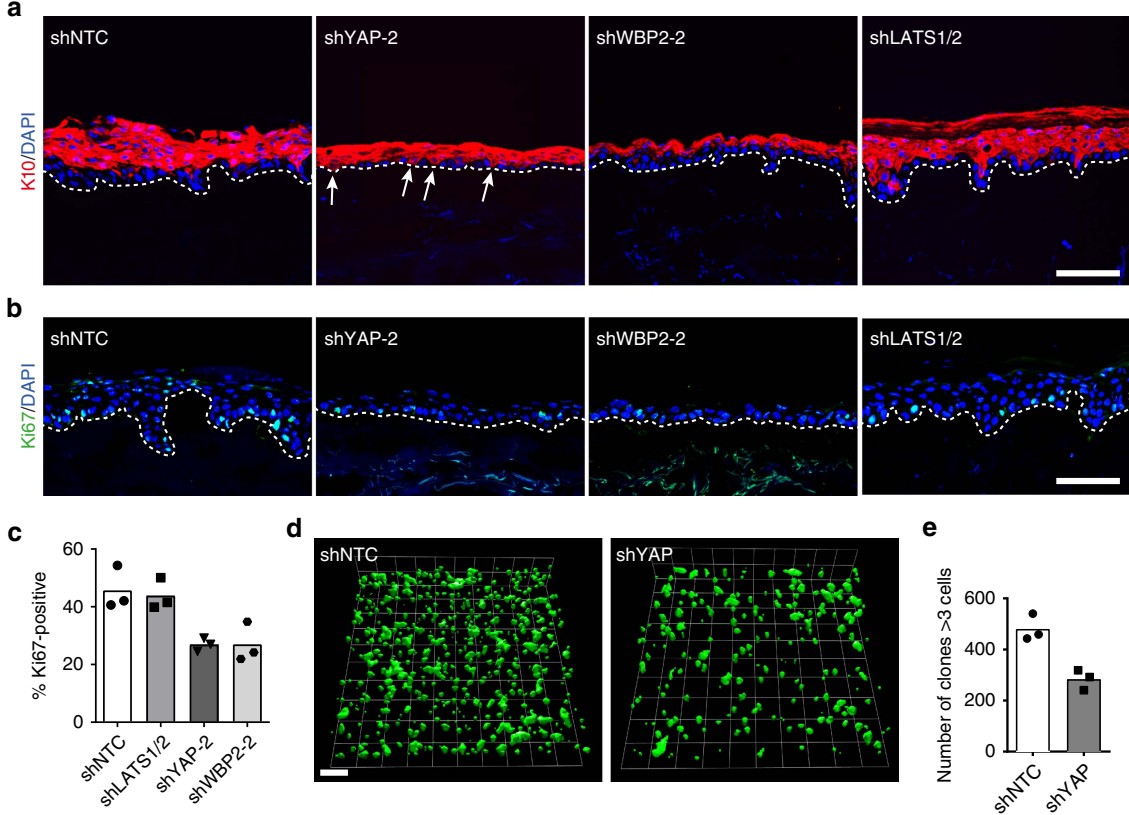

**Figure 6 | YAP and WBP2 are essential for reconstitution of human epidermis ex vivo. (a,b)** Representative images of human epidermis reconstituted by lentivirus-infected NHKs expressing either a non-targeting control shRNA (shNTC), a YAP-silencing shRNA (shYAP-2), a WBP2-silencing shRNA (shWBP2-2), or shRNAs silencing LATS1 and 2. Sections were immunolabelled with the indicated antibodies and counterstained with DAPI to reveal nuclei. White dotted lines demarcate dermal-epidermal boundaries. Arrows in (**a**) point to basal YAP knockdown cells displaying precococious expression of terminal differentiation-associated keratin(K)10. Scale bars, 100 μm. (**c**) Proliferative indices of resconstitued human epidermis. Data shown are from one independent experiment performed with three biological replicates (independent lentivirus infections and epidermal reconstitutions). Individual data points represent the average percentage of Ki67-positive basal epidermal cells from two immunolabelled sections representing different parts of the reconstituted epidermal tissue, bars represent the mean. (**d**) Three-dimensional rendering of whole-mount images of human epidermis reconstituted by mixtures of lentivirus-infected NHKs co-expressing a non-targeting control- (NTC) or a YAP-specific shRNA together with GFP, and uninfected cells. Scale bars, 150 μm. (**e**) Quantification of expanding GFP + clones (>3 cells) in reconstituted human epidermis. Data shown are from three independent experiments performed with six technical replicates (independent epidermal reconstitutions).

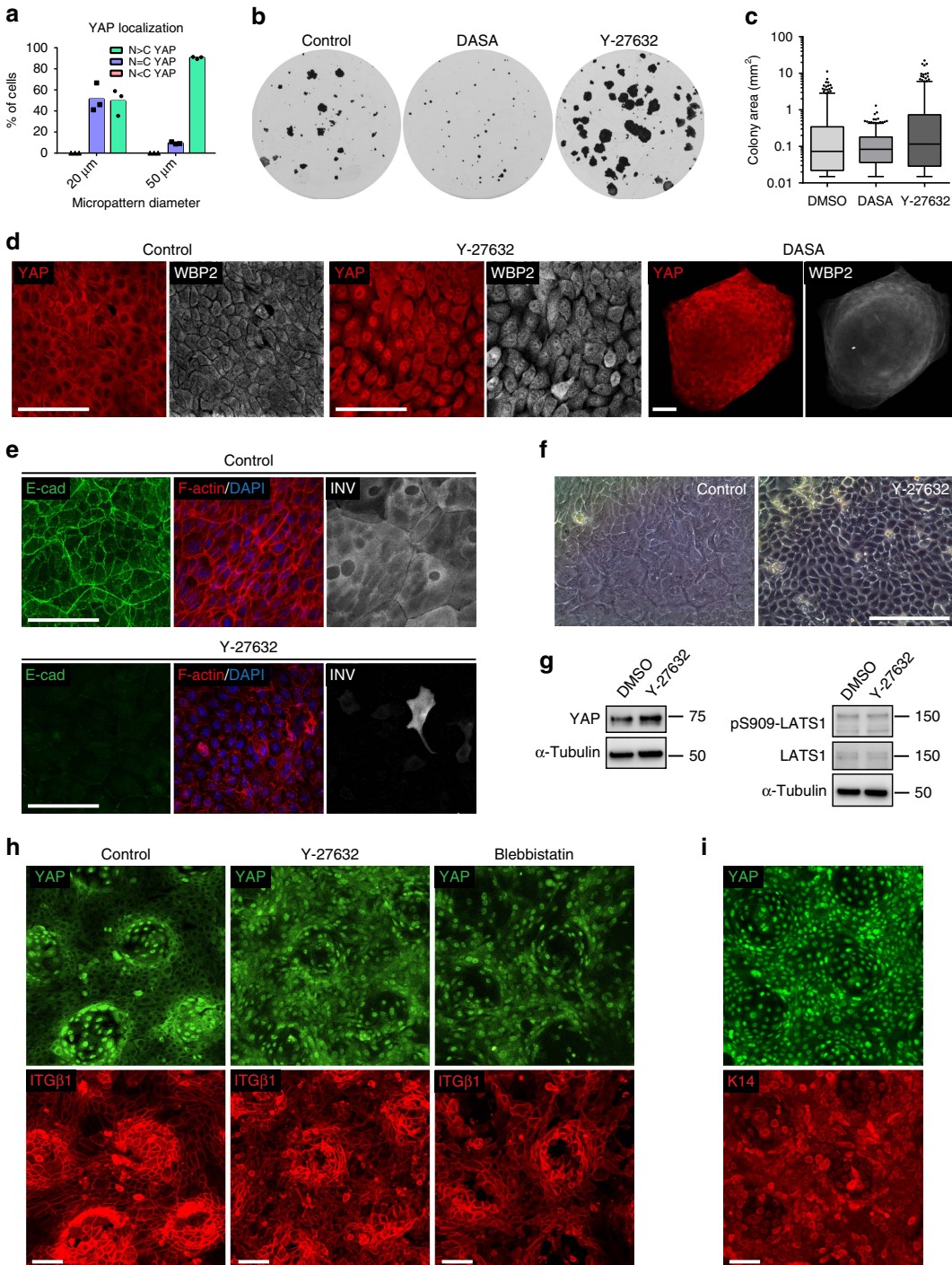

**Figure 7 | Regulation of YAP/WBP2 by contact inhibition.** (**a**) YAP immunofluorescence intensity was quantified in nuclear and cytoplasmic compartments by high content imaging analysis and cells were categorized according to their subcellular YAP localization pattern: N > C, predominantly nuclear YAP localization; N = C nuclear and cytoplasmic YAP localization; N < C, predominantly cytoplasmic YAP localization. Data shown are from three independent experiments. Individual data points represent the percentage of cells in each category (from >1,000 cells analysed per island size) in each experiment. Bars represent the mean. (**b–g**) NHKs were seeded at clonal density onto a fibroblast feeder layer, allowed to form expanding colonies over 5 days and then grown in the presence of Dasatinib (DASA, 4 μM) or Y-27632 (10 μM) for the last 9 days of culture. (**b**) Representative images of Crystal Violet-stained colonies. (**c**) Distribution of colony area. Box and whisker plots indicate the median (middle line in the box), 25th percentile (bottom line of the box), 75th percentile (top line of the box), and 5th and 95th percentiles (whiskers). 350 colonies were measured per condition. (**d–f**) Representative immunostainings (maximum intensity projections covering the basal cell layer) (**d,e**) using the indicated antibodies or DAPI and phalloidin to label nuclei and F-actin, respectively, and phase contrast images (**f**) of central areas of colonies. Scale bars, 100 μm. IVL, involucrin. (**g**) Cell lysates of colonies formed in the presence of Y-27632 (10 μM) or vehicle (DMSO) were analysed by western blotting using the indicated antibodies. Tubulin was used as loading control. (**h,i**) Representative images (maximum intensity projections) of NHKs (**h**) and SCC13 cells (**i**) grown on PDMS substrates with patterned topographical features in the presence of Y-27632 (10 μM) or Blebbistatin (50 μM), or vehicle (DMSO) alone (control), and immunolabelled with indicated antibodies and counterstained with DAPI to reveal nuclei. Scale bars, 100 μm.

differentiation. To test this, a GFP-IRES-shRNA lentiviral vector silencing YAP or a control shRNA vector were introduced into primary human keratinocytes 24 h before organotypic culture[15]. After transduction, ∼25–30% of keratinocytes expressed each GFP-IRES-shRNA construct (Supplementary Fig. 5h)[15]. Three-dimensional whole-mount imaging showed that by 3 weeks there were fewer YAP-targeted than control GFP-positive clones (Fig. 6d,e). Thus lineage tracing revealed that clonal expansion of basal cells was significantly impaired by YAP knockdown (Fig. 6d,e).

**Regulation of YAP/WBP2 by contact inibition**. Having established a role for YAP/WBP2 in regulating NHK growth, we next examined the extrinsic signals that regulate YAP. When single NHKs are seeded on 20 μm diameter micro-patterned islands spreading is restricted and the cells undergo terminal differentiation. However, on 50 μm diameter islands the cells spread and do not differentiate[50]. Consistent with this, on 50 μm islands the majority of cells had nuclear YAP (Fig. 7a). However, on 20 μm islands most cells had either nuclear YAP or both cytoplasmic and nuclear YAP (Fig. 7a); thus adhesion to a small extracellular matrix area was not sufficient to cause efficient cytoplasmic tranlocation of YAP.

The nuclear exclusion of YAP/WBP2 in the compacted cells in the centre of mature NHK colonies (Fig. 3g and Supplementary Fig. 8b) suggested that contact inhibition might play a role in regulating YAP[51,52]. Knockdown of LATS1/2 did not prevent nuclear exclusion of YAP and S127-phosphorylated YAP was detected in the nucleus of cells at the colony periphery even though they had high nuclear YAP (Supplementary Fig. 10a,b), indicating that canonical Hippo signalling was not involved.

Contact inhibition of proliferation involves a gradual compaction of cells that is dependent on cell–cell adhesion and mediated by mechanical compression through cellular crowding[51,52]. Consistent with computational simulations demonstrating that the rate of adherens junction turnover can be a major determinant of colony growth[53], we found that impairment of adherens junction homoeostasis by inhibition of Rho GTPase signalling[54] with the Rho kinase (ROCK) inhibitor Y-27632 promoted colony expansion (Fig. 7b,c). Conversely, stabilization of adherens junctions by inhibition of Src family kinases (SFKs)[55], using the dual Src/Abl kinase inhibitor Dasatinib, blocked colony expansion (Fig. 7b,c). Neither drug affected total colony forming efficiency (Supplementary Fig. 11a). In the densely packed Dasatinib-treated colonies YAP and WBP2 were excluded from the nucleus (Fig. 7d). In contrast, both proteins were predominantly nuclear in Y-27632-treated colonies, where compromised adherens junction maintenance, altered F-actin organization, reduced cell compaction and impaired stratification of terminally differentiated cells were evident (Fig. 7d–f). Y-27632-treatment increased the total level of YAP without affecting total and activated LATS1 levels (Fig. 7g), confirming that canonical Hippo signalling did not control YAP/WBP2 signalling in the context of contact inhibition. Dasatinib or Y-27632-treatment had no effect on nuclear YAP/WBP2 abundance when single NHKs were confined to micro-patterned adhesive islands (Supplementary Fig. 11b,c), arguing against a role of focal adhesion-transduced cytoskeletal tension in regulating YAP/WBP2 (ref. 56). However, when cell–cell contact was permitted on non-patterned substrates, Dasatinib readily induced adherens junctions formation, leading to micro-colony formation, cell compaction and nuclear YAP/WBP2 exclusion (Supplementary Fig. 11d–g). Y-27632-treatment alone had no such effect (Supplementary Fig. 11d,e,g), but could block Dasatinib-induced cell compaction and preserve nuclear

YAP/WBP2 localization (Supplementary Fig. 11g). Thus, cell compaction, promoted by stabilization of adherens junctions, controls the subcellular localization of YAP/WBP2 in NHKs.

We next tested if the same mechanism was also operating in the context of confluent epidermal cell sheets. Using PDMS substrates that mimic the topography of the epidermal-dermal interface to recreate the patterned distribution of SCs in adult human epidermis[14], we observed nuclear YAP and WBP2 localization to be restricted to integrin β1-bright SC clusters at the tips of the topographies (Supplementary Fig. 12a,b). As shown in Fig. 7h, this highly specific patterning of YAP activity could be efficiently disrupted by blocking signalling downstream of Rho GTPase with Y-27632 or the Non-Muscle Myosin II inhibitor Blebbistatin. Culturing NHKs under low Ca$^{++}$ conditions to prevent adherens junction formation also disrupted the patterning of nuclear YAP (Supplementary Fig. 12c). Nuclear YAP was observed in all SCC13 cells (Fig. 7i), consistent with their inability to form stable adherens junctions[41,57] as evidenced by reduced α-catenin expression (Supplementary Fig. 4a). We conclude that contact inhibition controls the subcellular localization of YAP/WBP2 in epidermal sheets.

## Discussion

Our genome-wide screens identified YAP/WBP2/TEAD as driving clonal expansion of NHK and cSCC SCs *in vitro*. While YAP and WBP2 cooperate to promote clonal expansion of normal and neoplastic human epidermal SCs by driving TEAD-mediated transcription of proliferation/survival-promoting genes, YAP supresses the onset of terminal differentiation independently of WBP2. Consequently, in cSCC cells constitutive nuclear localization of YAP/WBP2 contributes to their inability to undergo terminal differentiation.

Although YAP is by default a transcriptional co-activator, it can also act as a co-repressor in a complex with TEAD transcription factors and distinct chromatin-modifying proteins[58]. Our data suggest that YAP might act in cooperation with TEAD1 to repress terminal differentiation-associated genes, consistent with the observation that TEAD1 suppresses involucrin gene transcription[59]. The co-suppressor functions of YAP are likely contained within its C-terminus, as overexpression in mouse epidermis of a constitutively active YAP lacking the C-terminus does not block terminal differentiation[60]. The effect of WBP2 in enhancing the transcriptional co-activator functions of YAP likely involves the recruitment of factors that promote chromatin remodelling and productive transcriptional elongation[61]. In breast cancer cells, WBP2 cooperates with β-catenin and YAP/TAZ to drive TCF-mediated gene transcription downstream of WNT signalling[62]. Consistent with findings from other groups demonstrating that NHKs are only weakly responsive to WNT signals[63], our NHK screen did not reveal any involvement of WNT signalling in controlling YAP activity in human epidermal SCs and in driving their clonal expansion. Thus, different cell types appear to engage YAP/WBP2 via distinct signalling mechanisms for different transcriptional outputs.

The skin phenotype of WBP2 knockout mice largely phenocopied that of mice with a conditional, keratinocyte-specific knockout of YAP and/or TAZ[30,41,42]. Knockout of YAP alone results in reduced proliferation in the embryonic and neonatal epidermis, while YAP and TAZ have overlapping functions in promoting proliferation in the regenerating adult epidermis. However, contrary to the hair loss phenotype reported for one keratinocyte-specific YAP/TAZ knockout mouse line[42], WBP2 knockout mice had no obvious hair cycle abnormalities. Thus, WBP2's functions in mouse skin *in vivo* appear to be

restricted to the IFE, where WBP2 is important for promoting proliferation of actively cycling epidermal SCs during tissue development and regeneration.

In mouse epidermis YAP is positively regulated by an integrin-focal adhesion kinase-SFK signalling axis[42], and negatively regulated by cell–cell contact[41,57]. Our finding that nuclear YAP localization in single cells was unaffected by Dasatinib or Y-27632 treatment argues against an involvement of SFKs and ROCK in regulating YAP downstream of integrins. Similar observations were also made by Das et al.[64], who found that in the absence of cell–cell contact, nuclear localization of YAP is independent of integrin signalling and Myosin II contractility, but is promoted by the actin cytoskeleton. Our data suggest within epidermal sheets SFKs and ROCK regulate YAP by modulating the stability of adherens junctions and the organization and contractility of the associated actin cytoskeleton and thereby the extent of cell compaction, which is an essential cue for efficient contact inhibition[51]. Supporting our findings, hyperactivation of Src was found to drive breast cancer cell invasion by destabilizing adherens junctions and promoting nuclear localization of YAP[65]. Our findings are also consistent with those of Aragona et al.[51] who showed that mechanical forces transduced and imposed by the cytoskeleton are the overarching regulators of YAP/TAZ in multicellular contexts and control their activity independently of phosphorylation by LATS1/2. Together, our data suggest an important role for adherens junctions and the actin cytoskeleton in regulating the epidermal SC compartment and shed new light on the proliferation-promoting effect of Y-27632 on primary epidermal cell cultures[11,66].

Our findings provide an explanation for why cSCC cells with reduced α-catenin levels (and thus impaired adherens junction formation[41,57]) fail to exclude YAP/WBP2 from the nucleus at high cell densities and suggest a role of WBP2 in stabilizing YAP when cell–cell adhesion is low or absent. We hypothesize that when YAP is sequestered on 14-3-3 proteins and α-catenin upon contact inhibition[41], it is protected from phosphorylation-induced proteosomal degradation[29], but in the absence of stable cell–cell adhesion, such as is found in the regenerating epidermis or in cSCC, cells rely on WBP2 to stabilize YAP.

We conclude that YAP, in cooperation with WBP2, is a key driver of proliferation of actively cycling epidermal SCs in vitro and in vivo. Since YAP functions are preserved in other stratified epithelia[38,67,68], our findings may help to elucidate the molecular regulation of proliferative heterogeneity in these tissues.

## Methods

**Mice.** All animal experiments were subject to local ethical approval and performed under the terms of a UK government Home Office license. Wbp2[tm2a(EUCOMM)Wtsi] mice (outbred on a C57BL6/CBA background) were provided by Karen P. Steel[49]. Homozygous mutant mice are referred to as Wbp2[−/−]. Wild-type (Wbp2[+/+]) and heterozygous (Wbp2[+/−]) mice were used as controls and showed no apparent phenotype in skin[49] (Supplementary Fig. 9). Animals were genotyped using standard procedures, and with the recommended set of primers[49]. In the developmental analysis time course both male and female mice were analysed, while for the wound healing experiment only 8-weeks-old adult female mice were used. Before wounding, back skin hair was clipped and mice were anaesthetized using isofluorane (CP-Pharma) and injected subcutaneously with the analgesic Vetergesic (diluted 1:20 in sterile PBS; 100 µl per 20 g body weight; Ceva Animal Health). A 5 mm diameter biopsy punch (Stiefel) was used to introduce a full-thickness wound in the central back skin, and wounds were excised at the indicated time points and embedded in optimal cutting temperature compound (OCT). All tested animals were included; no statistical method was used to predetermine sample size; no randomization or blinding was used.

**Human tissues.** All human tissues were collected after informed consent and in compliance with either UK national regulations (UK Human Tissue Act (2004)) or German national regulations (German Medical Council). Human embryonic and fetal tissues were obtained with appropriate ethical approval from the UK Human Developmental Biology Resource (www.hdbr.org).

**Pharmacological inhibitors.** The ROCK inhibitor Y27632 was obtained from Enzo (ALX-270-333) and the Myosin II inhibitor Blebbistatin from Cambridge Bioscience (CAY13165). PP2 and IWP2 were from Tocris (#3533, #1407) and Dasatinib was from Cell Signalling Technologies (#9052). Y27632 was dissolved in sterile Milli-Q water; all other inhibitors were dissolved in sterile DMSO.

**Microfabrications.** CYTOOchips Arena micro-patterned glass slides and custom-manufactured composite micro-patterned glass slides containing arrays of 20 and 50 µm diameter circular islands were purchased from CYTOO (Grenoble, France); the custom-designed template masks to print new slides are available upon request (Custom_CC23_Q13-24-38). PDMS elastomer substrates mimicking the topography of the epidermal-dermal interface were fabricated as described[14]. Topography S1 was used for experiments[14].

**Cell culture.** Stock cultures of primary normal human keratinocytes (NHKs, strain km) were obtained from surgically discarded foreskin. SCC13 cells are an established cutaneous SCC cell line isolated from a cutaneous SCC of facial epidermis[10], and were obtained from Dr. James Rheinwald (Department of Dermatology, Harvard Skin Research Centre, USA), not authenticated. NHK and SCC13 cells were used in all experiments at passages 2–5. SCC-NR cells were isolated by Dr Nicholas Rabey at Addenbrooke's Hospital (Cambridge, UK) from a portion of a surgically excised cutaneous SCC from the back of the hand of an elderly male patient, and were used in all experiments at passages 1 or 2. HEK-293 T cells were purchased from ATCC (CRL-3216). 3T3-J2 fibroblasts were originally obtained from Dr James Rheinwald (Department of Dermatology, Harvard Skin Research Centre, USA), not authenticated. All cell stocks were routinely tested for mycoplasma contamination and were negative. None of the cell lines used in this study is present in the database of commonly misidentified cell lines. NHKs and cutaneous SCC cells were cultured in complete FAD medium, containing 1 part Ham's F12, 3 parts Dulbecco's modified eagle medium (DMEM), $10^{-4}$ M adenine, 10% (v/v) FBS, 0.5 µg ml$^{-1}$ hydrocortisone, 5 µg ml$^{-1}$ insulin, $10^{-10}$ M cholera toxin and 10 ng ml$^{-1}$ EGF, on mitotically inactivated 3T3-J2 cells as described previously[9,12]. HEK-293 T cells were cultured in high-glucose DMEM (Sigma-Aldrich) supplemented with 100 IU ml$^{-1}$ penicillin, 100 µg ml$^{-1}$ streptomycin and 10% (v/v) FBS (Life Technologies). 3T3-J2 cells were cultured in high-glucose DMEM (Sigma-Aldrich) supplemented with 100 IU ml$^{-1}$ penicillin, 100 µg ml$^{-1}$ streptomycin and 10% (v/v) adult BS (Life Technologies). 3T3-J2 cells were mitotically inactivated either by irradiation or by mitomycin C-treatment (3 h, 4 µg ml$^{-1}$ final concentration, Sigma-Aldrich). For experiments shown in Supplementary Fig. 11d–g, cells were seeded into cell culture multi-well plates (Falcon) or onto glass coverslips, coated with rat-tail collagen type I (20 µg ml$^{-1}$ in PBS, BD Biosciences), and grown in the absence of feeder cells in keratinocyte serum-free medium (KSFM) containing 30 µg ml$^{-1}$ bovine pituitary extract and 0.2 ng ml$^{-1}$ EGF (Thermo Fisher Scientific) for 24 h before treatment. For induction of terminal differentiation, cells were seeded at a density of $3.5 \times 10^4$ cells per cm$^2$ into cell culture dishes (Falcon) coated with rat-tail collagen type I (20 µg ml$^{-1}$ in PBS, BD Biosciences), grown to confluence in KSFM and then switched to complete FAD medium. For experiments using glass microchips containing micro-patterned arrays of 20 and/or 50 µm diameter circular islands, pre-confluent NHKs were gently disaggregated in trypsin/EDTA following removal of the feeder layer, filtered twice through a 40 µm cell strainer (Falcon) and re-seeded onto the substrates at a density of $1 \times 10^4$ cells cm$^{-2}$ in KSFM. Cells were allowed to adhere for 1 h and the substrates were then rinsed five times with fresh medium to remove non-attached cells. To generate micro-patterned arrays of micro-epidermis, NHKs or SCC13 cells were seeded onto CYTOOchips Arena at a density of $2 \times 10^5$ cells per cm$^2$ in complete FAD medium. After removing cells not adhering within 1 h by extensive washing, the remaining adherent cells were then cultured for 48 h. For the small-scale siRNA screen, $1 \times 10^5$ siRNA SMARTpool-transfected NHKs were seeded per well into 24-well cell culture plates (Falcon) coated with rat-tail collagen type I (20 µg ml$^{-1}$ in PBS, BD Biosciences), and cultured for 48 h in complete FAD medium. NHKs were seeded onto rat-tail collagen type I (20 µg ml$^{-1}$ in PBS, BD Biosciences)-coated PDMS substrates in complete FAD medium at a density of $5 \times 10^5$ cells per cm$^2$. After 45 min, substrates were rinsed gently to remove non-adherent cells and transferred to 12-well cell culture plates (Falcon) containing a 3T3-J2 feeder layer[14].

**Cell fractionation by differential adhesion.** Cell fractions enriched for human epidermal SCs, CP/transient amplifying cells and terminally differentiated cells were prepared as described[12], using human placenta collagen type IV (Sigma-Aldrich). For fluorescence microscopy of colonies, feeder cells were added at low density after SCs or CP/transit amplifying cells had been captured, and the cells were cultured for 10 days.

**Enrichment of cutaneous SCC-associated stem cells.** Enrichment of SCs from SCC13 cell cultures was achieved by spheroid growth-promoting culture on ultra-low adherence plates (Corning Costar) as described[69].

**Clonal growth assays.** 100 or 400–800 live (Trypan Blue-negative) NHK or SCC13 cells, respectively, were seeded per condition into duplicate or triplicate wells (containing a 3T3-J2 feeder layer) of a six-well dish (Falcon). After 12 days, feeder cells were completely removed by rinsing with PBS and colonies were either fixed in 4% (w/v) paraformaldehyde for 10 min and then stained with 1% Rho-danile Blue (1:1 mixture of Rhodamine B and Nile Blue A (Acros Organics))[12], or simultaneously fixed and stained with Crystal Violet solution (0.4% (w/v) crystal violet, 20% (v/v) methanol). Colonies were imaged and counted using an automated cell colony counter (Gelcount, Oxford Optronix, UK), and CFE was calculated as the average percentage of seeded cells that formed colonies. Colony area was measured using the Fiji image processing software package and the 'Analyse Particles' tools, with a minimum particle size of $0.01\,mm^2$.

**Epidermal reconstitution assays.** These were performed as described previously[15]. In brief, NHKs cultured on feeder cells to ∼70% confluence were disaggregated in trypsin/EDTA and cells ($5 \times 10^5$) were seeded into six-well cell culture plates (Falcon) coated with rat-tail collagen type I ($20\,\mu g\,ml^{-1}$ in PBS, BD Biosciences), and cultured for 24 h in KSFM medium. NHKs were then infected with MISSION lentiviral particles (Sigma-Aldrich; the sequences of the shRNAs can be found in Supplementary Table 1) at a multiplicity of infection (MOI) of two in the presence of $5\,\mu g\,ml^{-1}$ polybrene (Sigma-Aldrich). Medium was replaced after 24 h and shRNA-expressing cells were selected for 72 h using puromycin ($2\,\mu g\,ml^{-1}$, Sigma-Aldrich). At 96 h after infection, cells were collected and seeded ($5 \times 10^5$) on irradiated de-epidermized human dermis in six-well Transwell plates with feeder cells and cultured at the air–liquid interface for three weeks[15]. For analysis of clonal expansion during epidermal reconstitution, we used pGIPZ-based lentiviral vectors (GE Dharmacon) expressing miR30 embedded shRNAs silencing YAP (Clone V3LHS_306099) or a non-targeting control shRNA (#RHS4346). GFP was expressed from the same promoter through an IRES. Three-dimensional imaging of reconstituted epidermis was performed on a two-photon Leica confocal microscope. Quantification of the unprocessed data was performed using Volocity software.

**WBP2 overexpression.** Pre-confluent NHK cultures were disaggregated and cells ($2 \times 10^5$) were seeded into six-well cell culture plates (Falcon) coated with rat-tail collagen type I ($20\,\mu g\,ml^{-1}$ in PBS, BD Biosciences), and cultured for 24 h in KSFM medium. NHKs were then infected with Precision LentiORF (GE Dharmacon) lentiviral particles co-expressing different WBP2 ORFs (OHS5901, OHS5902) or TurboRFP and nuclear localized TurboGFP as fluorescent reporter at an MOI of 5 in the presence of $5\,\mu g\,ml^{-1}$ polybrene (Sigma-Aldrich). Medium was replaced after 24 h and cells were harvested after 72 h. Quantification of TurboGFP expression revealed transduction efficiencies of ∼70%.

**Genome-wide pooled shRNA screens.** Details can be found in Supplementary Methods.

**RNA extraction and real-time qPCR.** Total RNA was isolated from cultured cells using the RNeasy kit (Qiagen). Complementary DNA was generated using the QuantiTect Reverse Transcription kit (Qiagen). Quantitative PCR (qPCR) analysis of cDNA was performed using qPCR primers (published or designed with Primer3) and Fast SYBR green Master Mix (Life Technologies). RT-qPCR reactions were run on CFX384 Real-Time System (Bio-Rad). 18S rRNA was used as a housekeeping gene for normalization. Please refer to Supplementary Table 2 for qPCR oligos sequences.

**Plasmid DNA transfection.** HEK-293 T cells were transfected with plasmid DNAs in six-well dishes (Falcon) using jetPRIME reagent (Polyplus Transfection) following the manufacturer's instructions.

**siRNA transfection.** siRNA nucleofection was performed with the Amaxa 96-well shuttle system (Lonza). Pre-confluent NHK cultures were disaggregated and resuspended in cell line buffer SF (Lonza). For each $20\,\mu l$ transfection (program FF-113) reaction, $2 \times 10^5$ cells were mixed with $1\,\mu M$ siRNA duplexes as described previously[15]. Transfected cells were allowed to recover at ambient temperature for 10 min and were subsequently re-plated into rat-tail type I collagen ($20\,\mu g\,ml^{-1}$ in PBS, BD Biosciences)-coated cell culture well plates (Falcon) and, unless stated otherwise, grown in KSFM medium for 24 h before being used for downstream assays. SMARTpool ON-TARGET plus siRNAs (GE Dharmacon) were used for gene knockdown. Each SMARTpool was a mix of four sets of RNAi oligos (the sequences of siRNA oligos can be found in Supplementary Table 3). Non-targeting control siRNAs were from Ambion (AM4611 and AM4637). SCC13 cells ($5 \times 10^5$) were reverse transfected with siRNAs (25 pmol) in six-well dishes (Falcon) using Lipofectamine RNAiMAX reagent in combination with Opti-MEM medium (Life Technologies) according to the manufacturer's instructions.

**Chemiluminescent reporter gene assays.** Reporter gene assays were performed using the Dual-Light System (Applied Biosystems). ∼50% confluent HEK-293 T

were transfected with SMARTpool siRNAs (12 pmoles $cm^{-2}$) in six-well cell culture plates (Falcon) using jetPRIME reagent (Polyplus Transfection) following the manufacturer's instructions. Twenty-four hours after siRNA transfection, TEAD reporter plasmid (8xGTIIC–LUX[56]; a gift from Stefano Piccolo, University of Padova, Italy; Addgene #34615) was transfected into the cells (200 ng $cm^{-2}$) together with β-galactosidase plasmid (pEF-1α-LacZ 200 ng $cm^{-2}$, Addgene #17430; kindly provided by Dr Shukry Habib, CSCRM, King's College London, UK) to normalize for transfection efficiency, using jetPRIME reagent (Polyplus Transfection) following the manufacturer's instructions. At 24 h post-plasmid transfection, cells were trypsinized and re-plated ($1 \times 10^4$ cells) in triplicates into a 96-well plate, and allowed to grow for a further 24 h. Relative luciferase units were measured on a Glo-Max-Multi+ Multimode Reader (Promega) and normalized against β-galactosidase activity.

**Immunofluorescence microscopy of cultured cells.** Cultured cells grown on CYTOO microchips or on glass coverslips (coated with rat-tail type I collagen ($20\,\mu g\,ml^{-1}$ in PBS, BD Biosciences)) were fixed in 4% (w/v) paraformaldehyde (Sigma) for 10 min and permeabilized with 0.2% (v/v) Triton X-100 for 5 min at ambient temperature. NHK colonies grown in the presence of a 3T3-J2 feeder layer were fixed in 4% (w/v) paraformaldehyde (Sigma) for 45 min and permeabilized with 0.5% (v/v) Triton X-100 for 45 min at ambient temperature. Confluent NHK sheets grown on PDMS substrates with patterned topographies were processed for immunofluorescence microscopy as previously described[14]. Samples were blocked for 1 h in 10% (v/v) FBS plus 0.25% (v/v) fish skin gelatin (Sigma-Aldrich) in 1× PBS (blocking buffer), and incubated with primary antibodies (diluted in blocking buffer) in a humid chamber overnight at 4 °C. After washing with PBS, samples were incubated with Alexa Fluor-conjugated secondary antibodies for 2 h at room temperature. Primary antibodies are listed in Supplementary Data 13. Rhodamine–phalloidin (Thermo Fisher Scientific) was included in the secondary antibody solution where indicated. Fixed and stained coverslips were mounted on glass slides with ProLong Gold anti-fade reagent with DAPI (Thermo Fisher Scientific). Confocal images were acquired with a Nikon A1plus or A1r point scanning confocal fluorescence microscope equipped with Plan Fluor ELWD 20x Ph2 ADL-, S Plan Fluor ELWD 20x Ph1 ADL-, Plan Apo VC 20x DIC N2-, Plan Apo 40x DIC M N2-, or Plan Fluor ELWD 40x Ph2 ADL objectives (Nikon) and controlled by NIS elements C software. Digital images were processed using NIS elements Advanced Research and Adobe software packages.

***In situ* proximity ligation assays.** *In situ* PLA was performed with DuoLink In Situ Reagents from Olink Bioscience (Sigma-Aldrich). NHK cells were seeded onto glass coverslips coated with rat-tail type I collagen ($20\,\mu g\,ml^{-1}$ in PBS, BD Biosciences), grown for 24 h in complete FAD medium, and fixed in 4% (w/v) paraformaldehyde (Sigma) for 10 min at room temperature. PLAs were performed as indicated by the provider's protocol, after an overnight incubation with primary antibodies following the immunofluorescence protocol described above. Primary antibodies are listed in Supplementary Data 13.

**Immunofluorescence microscopy of tissue sections.** Human and mouse tissue samples were embedded in OCT compound and $12\,\mu m$ sections (mouse and human skin) or $10\,\mu m$ sections (reconstituted human skin) were cut on a cryostat and fixed in 4% (v/v) paraformaldehyde (Sigma). Tissues obtained from skin biopsies of cancer patients or healthy donors were fixed in 4% (w/v) paraformaldehyde (Sigma) for 24 h and embedded in paraffin after dehydration. $5\,\mu m$ thick sections were deparaffinised, subjected to antigen retrieval using citrate buffer, and permeabilized with 0.5% (v/v) Triton X100 in PBS. Tissue sections were incubated in blocking buffer (3% (v/v) BSA, 5% (v/v) FBS, 0.25% (v/v) fish skin gelatin, in 1× PBS) for 20 min before staining with primary antibodies overnight at 4 °C. In some cases, primary antibodies had been directly conjugated with Alexa fluorophores using Molecular Probes Antibody Labelling Kits (Thermo Fisher Scientific). Otherwise, Alexa Fluor-conjugated secondary antibodies were used. Primary antibodies are listed in Supplementary Data 13. DAPI was used to label nuclei. Confocal images were acquired with a Nikon A1plus point scanning confocal fluorescence microscope equipped with a CFI Plan Apo λ ×40 objective (Nikon) and controlled by NIS elements C software. Digital images (single optical sections) were processed using NIS elements Advanced Research and Adobe software packages.

**High content imaging analysis.** Cells were processed for immunofluorescence microscopy as described above. Automated image acquisition was performed on an Operetta high content imaging system (Perkin Elmer). For image scanning, 25–30 fields (quantification of nuclear YAP abundance, Supplementary Figs 5n and 11c,d), 60–80 fields (quantification of subcellular YAP localization, Fig. 7a), or 150–200 fields (quantification of PLA signals, Fig. 2e,f,h and Supplementary Figs 5m and 11f) without fluorescence artifacts were selected for each glass coverslip or CYTOO chip, and z-stack images (z-dimension $1\,\mu m$) were acquired using ×10 (quantification of nuclear YAP abundance, Supplementary Figs 5n and 11c,d; quantification of subcellular YAP localization, Fig. 7a) and ×20 (quantification of PLA signals, Fig. 2e,f,h and Supplementary Figs. 5m and 11f) long WD objectives via DAPI (20 ms exposure), Alexa-488 (200 ms exposure) and Alexa-594 (200 ms

exposure) channels. Images were analysed using custom algorithms in the Harmony high content analysis software package (Perkin-Elmer). For all image analyses, cells were initially defined using the DAPI channel, then the cytoplasm was segmented using one of the Alexa channels. Nuclear abundance was calculated as the ratio of median fluorescence intensity in the nucleus to median fluorescence intensity in the cytoplasm. The complete Harmony image analysis sequences can be provided upon request.

**Semi-quantitative analysis of immunofluorescence intensities.** Immuno-fluorescence intensities in sections of normal human skin and from cSCCs were scored by one dermatologist (S.Q.) and one biologist (B.L.), who was blinded to clinical outcome, using the H-score method[70]. Cellular YAP and WBP2 immunofluorescence intensities were first scored as 0, 1, 2 or 3 corresponding to the presence of no signal in the basal layer signal in normal tissue, signal intensity similar to basal layer signal in normal tissue, signal intensity higher than basal layer signal in normal tissue, and signal intensity much higher relative to basal layer signal in normal tissue. The average percentage of positive cells was then calculated and the following formula applied: H-score = (% of cells stained at intensity category $1 \times 1$) + (% of cells stained at intensity category $2 \times 2$) + (% of cells stained at intensity category $3 \times 3$). An H-score between 0 and 300 was obtained where 300 was equal to 100% of cells stained strongly $(3+)$.

**Bioinformatics analyses.** Details can be found in Supplementary Methods.

**Western blotting.** Cells were lysed on ice for 30 min in $1 \times$ RIPA buffer (Cell Signalling Technology) supplemented with PhosSTOP Phosphatase Inhibitor and cOmplete EDTA-free Protease Inhibitor Cocktails (Roche), and RIPA-soluble and -insoluble proteins were separated via centrifugation ($16,000 \times g$ for 20 min at 4 °C). Total protein amount was quantified in RIPA extracts using the BCA kit (Pierce). Equivalent quantities of RIPA-solubilized proteins were resolved by SDS-PAGE in 4–20% Criterion TGX Stain-Free Precast Gels and transferred to Immun-Blot Low Fluorescence polyvinylidene difluoride membranes (Bio-Rad Laboratories) using the Trans-Blot Turbo transfer system (Bio-Rad Laboratories). Protein transfer and equal protein loading were confirmed by enhanced tryptophan fluorescence imaging of polyvinylidene difluoride membranes (Bio-Rad Laboratories). Membranes were blocked with either 5% (w/v) non-fat milk or 5% (w/v) BSA in PBS supplemented with 0.05% (v/v) Tween-20 (PBS-T) and then probed with the indicated antibodies diluted in blocking buffer. Primary antibody-probed blots were visualized with appropriate horseradish peroxidase-coupled secondary antibodies (Jackson ImmunoResearch) using enhanced chemiluminescence (Clarity Western ECL, Bio-Rad Laboratories) according to the manufacturer's instructions. Protein bands were detected using a ChemiDoc Touch Imaging System (Bio-Rad Laboratories). Processing of western blot images was done by Image Lab software (Bio-Rad Laboratories). For quantifications of band intensities, exposures within the dynamic range were chosen. Images of uncropped blots are shown in Supplementary Fig. 13.

**Immunoprecipitation.** For co-immunoprecipitations of endogenous protein complexes, cells were lysed on ice for 45 min in 50 mM Tris-HCl (pH 8.0), 150 mM NaCl, 10% (v/v) glycerol, 0.5% (v/v) Triton X-100, 1 mM MgCl$_2$ and protease and phosphatase inhibitors. Equal quantities of solubilized proteins (500 µg ml$^{-1}$) were incubated with anti-YAP- or anti-WBP2 antibodies overnight at 4 °C. Immunoprecipitates were captured on protein G-agarose columns (Active Motif), washed three times with NET buffer (50 mM Tris-HCl (pH 7.5), 150 mM NaCl, 1 mM EDTA, 0.25% (v/v) gelatine and 0.1% (v/v) Nonidet P-40) and analysed by immunoblotting.

**Reproducibility of experiments.** For genome-wide pooled shRNA screens, two independent experiments were performed. For the small-scale siRNA screen, five independent experiments were performed with two biological replicates (independent siRNA transfections). For all other experiments involving pharmacological inhibitor-, siRNA- or shRNA treatments, three to five independent experiments were performed with two biological replicates (independent siRNA or shRNA transfections or drug treatments) per condition. For epidermal reconstitution assays, one experiment was performed with three biological replicates (independent lentivirus infections and epidermal reconstitutions). For quantification of expanding GFP + clones in reconstituted human epidermis, three independent experiments were performed with six technical replicates. For clonal growth assays two to three independent experiments were performed with two to three technical replicates per condition. Western blots, PLA assays, and high content imaging-based quantifications of immunostainings were performed at least twice with similar results. For luciferase reporter assays, each experiment contained three biological replicates (independent siRNA transfections) and was repeated twice independently with similar results. For immunostainings, representative images from one out of three experiments are shown. For immunoprecipitations two independent experiments were performed with similar results.

**Statistics and graph generation.** No statistical method was used to predetermine sample size. Data sets with sufficient $n$ numbers were first analysed using the D'Agnostino & Pearson Omnibus normality test. In cases where the data were not found to be normally distributed, they was subsequently analysed using appropriate non-parametric tests (specified in figure legends), provided that the variance was comparable between the groups analysed. In cases of small data sets ($n < 3$), the individual data points are displayed in the graphs. All graphs were generated using GraphPad Prism 6 and 7.

**Antibodies.** Primary and secondary antibodies are listed in Supplementary Data 13.

**Data availability.** The authors declare that all data supporting the findings of this study are available within the paper and its Supplementary information files. Raw Illumina sequencing data from the genome-wide pooled shRNA screens are deposited in the Gene Expression Omnibus (GEO) under the accession number GSE79560.

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

## Acknowledgements

We gratefully acknowledge the financial support of the Wellcome Trust and Medical Research Council and funding from the Department of Health via the National Institute for Health Research comprehensive Biomedical Research Centre award to Guy's & St Thomas' National Health Service Foundation Trust in partnership with King's College London and King's College Hospital NHS Foundation Trust. G.W. was the recipient of an EU Marie Curie Intra-European Fellowship (FP7-PEOPLE-2012-IEF-326619. E.R. was the recipient of a European Molecular Biology Organization (EMBO) long-term fellowship (ALTF594-2014). B.M.L. was the recipient of a Federation of European Biochemical Societies (FEBS) long-term fellowship. We thank Professor Jason A. Burdick (University of Pennsylvania) and Murat Guvendiren (Rutgers University) for providing substrates with patterned topographies. We thank Karen P. Steel (King's College London) for providing WBP2 knockout mice, and Annalisa Buniello (King's College London) for mouse genotyping. We thank Petra Boukamp, Elizabeth Pavez Lorie and Eileen Gentleman for critical discussions, Simon Broad and Nicholas Rabey for providing primary human cells and Davide Danovi for training in high content imaging.

## Author contributions

G.W. and S.W. conceived the study. S.W. designed and carried out the genome-wide RNAi screens. G.W. designed, carried out, analysed and interpreted the majority of in vitro experiments, prepared data for publication, and co-wrote the manuscript. E.R. and K.L.-A. designed and carried out animal experiments. A.O.P. performed bioinformatics analysis. B.M.L. carried out immunostainings of human normal skin and cancer tissue sections and performed semi-quantitative analysis of immunofluorescence intensities. A.M. performed siRNA experiments and qPCR analyses. S.B.T. performed immunostainings of human foetal and adult skin sections. P.V. performed immunostainings of cell sheets on substrates with patterned topographies. M.L. and L.M.R. performed western blotting experiments and clonal growth assays. G.D. helped with the genome-wide shRNA screens. S.Q. contributed normal and cancer tissue sections and performed semi-quantitative analysis of immunofluorescence intensities in tissue sections. F.M.W. consulted on experimental design and interpretation and co-wrote the manuscript.

## Additional information

**Competing interests:** The authors declare no competing financial interests.

