## [Peer Review File · Nature Communications]

Reviewers' comments:

Reviewer #1 (Remarks to the Author):

The authors conducted a novel and unbiased siRNA in vitro screen to identify molecules that support epidermal stem cell growth and maintenance. Using negative selection criteria, the investigators identify YAP-TEAD complex as a critical driver of stem cell expansion. Additionally, they connect WBP2 as an interacting partner and positive regulator of YAP-TEAD function. While the message of the manuscript is, overall, interesting, and strengthens the evidence of YAP as an epidermal stem cell regulator, the biochemical evidence and mechanism for YAP/TEAD interaction with WBP2 is weak.

1. The investigators identify WBP2 as an interacting molecule with the YAP-TEAD complex using proximity ligation assay (PLA). However, they do not provide any additional experimental evidence demonstrating the existence of WBP2-YAP-TEAD complex in epidermal stem cells (i.e. co-IP, ChIP-seq, in vitro protein interaction). Does WBP2 stabilize YAP-TEAD interaction? Does WBP2 depletion impact YAP-TEAD interaction in the nucleus? YAP-TEAD co-IP should be done in the context of WBP2 knockdown.
2. The siRNA experiments show that WBP2 depletion closely phenocopies YAP knockdown, but neglect to carefully assay epistatic interaction between YAP and WBP2. It is unclear whether WBP2 signals parallel with Hippo pathway or whether its interaction with YAP/TEAD directly impacts YAP transcriptional output. The investigators should knockdown WBP2 and introduce YAP cDNA (S94A, S127A mutants, and TEAD-YAP fusion) to assay for changes in Yap-target gene expression. The converse experiment using YAP knockdown with WBP2 overexpression should be done as well.
3. Although in vitro experiments provide direct evidence that YAP and WBP2 regulate epidermal stem cell expansion, they do not conduct any in vivo studies to confirm their findings. In fact, they state that several studies have suggested dispensability of YAP/TAZ in adult epidermal homeostasis. Also from our experience and many published papers, YAP/TAZ appear to be specifically required for cell growth in vitro while are completely dispensable for the corresponding cell type in vivo. Thus it is possible that described observations in the manuscript are specific to in vitro cell culture. Ideally, the authors should test their hypothesis using mouse models or grafting human epidermal cells in mouse recipients. The experiment should test whether depletion of YAP or WBP2 in the established (i.e. grafted or adult mouse epidermis) epidermis has an effect on homeostatic regeneration and differentiation.

Reviewer #2 (Remarks to the Author):

This is an outstanding manuscript.

The authors perform a large scale shRNA screen to identify key regulators of skin stem cell proliferation and SCC formation. They identify YAP and its co-factor WBP2, and proceed to confirm that they interact in the nucleus and function together to drive TEAD-dependent transcription. They further show that the two proteins are co-expressed in stem/progenitor populations in mouse skin and function to promote stem cell proliferation. They then investigate regulation of these factors by Integrin-Src and Rho-kinase signalling in the context of clonal expansion assays and their ability to drive stem cell proliferation.

These findings are novel, as analysis of YAP and WBP2 function in stem cell clonal expansion in skin and SCCs has not been previously performed. The results are therefore an important extension over the existing literature. The quality of the data is very high, with excellent statistical analysis and solidly drawn conclusions.

I have one minor comment that is not essential to address in order for this manuscript to be published.

Figure 5e is a striking result, because it shows that YAP can still be nuclear even when the Src inhibitor Dasatinib is applied, as long as the cell remains flat and basally anchored. This would suggest that a signalling mechanism other than Integrin-Src also operates to sense force/cell shape, which is an important conclusion. The authors further suggest that the effects of Dasatinib might be mediated by stabilisation of AJs that form between groups of cells (Fig S7e), which is an interesting hypothesis (and the reverse of the recently published a-catenin knockout mouse skin:

α E-catenin inhibits a Src-YAP1 oncogenic module that couples tyrosine kinases and the effector of Hippo signaling pathway.

Li P, Silvis MR, Honaker Y, Lien WH, Arron ST, Vasioukhin V.

Genes Dev. 2016 Apr 1;30(7):798-811. doi: 10.1101/gad.274951.115. Epub 2016 Mar 24.)

I am just curious why 2uM Dasatinib was used for these experiments (5e, S7e), rather than the 4uM in the previous experiment (5d), as the effects on YAP localisation in S7e are quite mild. Perhaps this experiment could be repeated with 4 or 5uM Dasatinib for 4 hours, rather than for 24 hours, to test whether YAP localisation might then be affected. It is a little surprising that you can treat NHK cells for 24 hours with these drugs at what is supposedly a high concentration without affecting cell morphology or viability - could the drug become partially degraded by the 24 hour time point?

Reviewer #3 (Remarks to the Author):

In this manuscript, Walko and colleagues report a role of YAP/WBP2 in conferring growth advantage to human epidermal stem cells and in their self-renewal activity. The factors were identified via an extensive genetic screening using normal and neoplastic epidermal cells. While both YAP and WBP2 were shown to be important for clonal expansion, only the former was capable of suppressing onset of terminal differentiation. Mechanistically, contact

inhibition acting via ROCK impinged upon Hippo signaling to suppress YAP/WBP2. Therefore, defective contact inhibition leads to unrestrained nuclear YAP/WBP2, leading to uncontrolled growth.

The fact that YAP/WBP conveys pro-growth properties is by itself not profoundly novel because there were papers showing that WBP2 homolog in Fly is a cofactor of Yorkie(YAP homolog) and YAP/TAZ plays a role in promoting SCCs growth.

However, I think that the present data and evidence regarding the role of YAP/WBP in human epithelial stem cell growth is still novel and very informative to be sufficiently published in Nat Comm. An unbiased extensive genetic screening approach and immunostaining results extensively revealed the role of YAP/WBP in clonal expansion and differentiation of epidermal cells. In particular, it is important that contact inhibition mediated by stabilization of adherent junction via Rho kinase worked together with Hippo signaling to inhibit YAP/WBP2.

I found that some data seem to be difficult to reconcile with, both as they are and also in light of published findings. To improve the quality of this manuscript, the following are comments I would like to suggest.

1) Having identified YAP/TAZ as key players in conferring growth advantage in epidermal cells, the authors simply assume TEAD dependency and denote future target gene analysis as 'YAP/TAZ/TEAD' or 'YAP/WBP2/TEAD'-dependent. However, the status of TEAD1/2/3/4 themselves have not been shown from the results of the genetic screen. If TEADs were not one of the highest hits comparable to YAP/TAZ/WBP2, the authors need to explain why this may be the case. I guess this may be due to multiple TEADs in cells. A related comment is to complement some key assays (i.e. colony formation, qPCR for differentiation markers) with TEADs knockdown.

2) The results in Fig.2d-I had been illustrated in other studies (Chan et al, Oncogene 2011; Zhang et al, Cell Death & Differentiation 2011) that YAP and WBP2 physically interact and the latter enhances the activity of the former. A more interesting question in the reviewer's opinion is whether the association between YAP/TEAD-WBP2 is increased in SCCs compared to normal epidermal cells or under conditions in which clonal growth is preferred (e.g. in Y27632-treated conditions, at least according to this manuscript; conversely, is this interaction decreased in abortive colonies comprising terminally-differentiated cells?).

3) Contact inhibition-related experiments presented in this study do not seem to be in line with the current understanding of mechanotransduction operating in the Hippo pathway. Although dasatinib has been shown to be a negative regulator of YAP (Rosenbluh et al, Cell 2012), Y27632 has also been shown to be a negative regulator of YAP by promoting YAP cytoplasmic translocation in the skin (Elbediwy et al, Development 2016). However, the current manuscript shows that Y27632 rather promotes clonal growth. The authors should therefore describe this discrepancy, and show why such an opposite phenomenon occurs. Perhaps it is because in this particular context YAP/WBP2 are not profoundly affected by canonical Hippo signaling including LATS kinases, but the authors nevertheless need to

clarify this point (i.e. most simply, checking whether Y27632 treatment does not reduce YAP/TEAD target genes).

A related question is, although the authors show LATS-induced YAP phosphorylation plays a minor role in functionally regulating the localization and activity of YAP, the authors claimed that the subcellular localization of YAP/WBP2 changes in response to adherens junction stabilization (i.e. Fig. 5g). What could be mediating such nucleo-cytoplasmic translocation irrespective of LATS?

4) The authors claim that they found no evidence of canonical Hippo signaling in regulation of clonal epidermal stem cell growth (Fig. 1j, S3g), since knock down of LATS1 or 2 did not increase YAP transcriptional output. However, it is known that LATS 1 and 2 are functionally redundant in some organ for example in the liver. I guess it may also be true in skin. So, it seems hasty to exclude the role of LATS in regulating YAP via depletion of one upstream negative regulator alone.

Besides, in Fig. S3f, the level of LATS1 protein seems to correlate with that of p-YAP (S127) (and inversely correlated with total YAP), implying a regulatory role of LATS in YAP activity in this context. So, unless authors knock down both LATS1/2 in this setting, authors can not rule out a possibility that canonical Hippo signaling including LATS1/2 affect YAP functions in cell growth.

It is note that canonical Hippo signaling including LATS1/2 may affect YAP function in cell differentiation but not in growth. In fact, mice lacking Sav1 (WW45) showed that mutant skin had a an expansion of Stem/progenitor cells with less differentiation (Lee et al, EMBO J. 2008 27(8):1231-42. This paper showed that Hippo pathway clearly is activated to inhibit YAP function during epithelial differentiation. Based on these, authors need to include this possible role of canonical Hippo signaling in regulation of epidermal stem cell growth or differentiation.

Response to referees

Reviewer #1

The authors conducted a novel and unbiased siRNA in vitro screen to identify molecules that support epidermal stem cell growth and maintenance. Using negative selection criteria, the investigators identify YAP-TEAD complex as a critical driver of stem cell expansion. Additionally, they connect WBP2 as an interacting partner and positive regulator of YAP-TEAD function. While the message of the manuscript is, overall, interesting, and strengthens the evidence of YAP as an epidermal stem cell regulator, the biochemical evidence and mechanism for YAP/TEAD interaction with WBP2 is weak.

1. The investigators identify WBP2 as an interacting molecule with the YAP-TEAD complex using proximity ligation assay (PLA). However, they do not provide any additional experimental evidence demonstrating the existence of WBP2-YAP-TEAD complex in epidermal stem cells (i.e. co-IP, ChIP-seq, in vitro protein interaction). Does WBP2 stabilize YAP-TEAD interaction? Does WBP2 depletion impact YAP-TEAD interaction in the nucleus? YAP-TEAD co-IP should be done in the context of WBP2 knockdown.

We have now confirmed the existence of a WBP2-YAP complex by immunoprecipitation in NHKs (Figure 2g). Although we could not co-immunoprecipitate TEADs, most likely due to their low abundance, we have included new data confirming the role of TEADs downstream of YAP and WBP2 (see below). We found no evidence that knock-down of WBP2 destabilized the interaction of YAP with TEAD transcription factors in the nucleus (Supplementary Figure 4j-l).

2. The siRNA experiments show that WBP2 depletion closely phenocopies YAP knockdown, but neglect to carefully assay epistatic interaction between YAP and WBP2. It is unclear whether WBP2 signals parallel with Hippo pathway or whether its interaction with YAP/TEAD directly impacts YAP transcriptional output. The investigators should knockdown WBP2 and introduce YAP cDNA (S94A, S127A mutants, and TEAD-YAP fusion) to assay for changes in YAP-target gene expression. The converse experiment using YAP knockdown with WBP2 overexpression should be done as well.

By overexpressing WBP2 in NHKs with reduced endogenous YAP levels we showed that WBP2's ability to enhance TEAD-mediated transcription of CYR61 depends on YAP. These new data are shown in the revised Figure 2l. This strengthens the conclusion from the luciferase reporter assays in Figure 2j and the overexpression experiments in Figures 2k that the interaction of WBP2 with YAP directly impacts on YAP/TEAD-mediated transcription.

3. Although in vitro experiments provide direct evidence that YAP and WBP2 regulate epidermal stem cell expansion, they do not conduct any in vivo studies to confirm their findings. In fact, they state that several studies have suggested dispensability of YAP/TAZ in adult epidermal homeostasis. Also from our experience and many published papers, YAP/TAZ appear to be specifically required for cell growth in vitro while are completely dispensable for the corresponding cell type in vivo. Thus it is possible that described

observations in the manuscript are specific to in vitro cell culture. Ideally, the authors should test their hypothesis using mouse models or grafting human epidermal cells in mouse recipients. The experiment should test whether depletion of YAP or WBP2 in the established (i.e. grafted or adult mouse epidermis) epidermis has an effect on homeostatic regeneration and differentiation.

We have now examined the skin of WBP2 knock-out mice and show that it largely phenocopies mice with a conditional, epidermis-specific knock-out of YAP and/or TAZ (Zanconato, F. et al. *Nature Cell Biology* 17, 1218-1227 (2015); Schlegelmilch, K. et al. *Cell* 144, 782-795 (2011); Elbediwy, A. et al. *Development* 143, 1674-1687 (2016)). WBP2 knock-out skin exhibits reduced proliferation in early post-natal epidermis and in adult epidermis during wound healing. We also found reduced nuclear YAP expression in regenerating WBP2-null epidermis, suggesting an additional role of WBP2 in modulating YAP protein levels in vivo. The new data are shown in Figure 4 and Supplementary Figures 5 and 8.

For completeness, we have now shown that WBP2 knockdown impairs human epidermal reconstitution in organotypic cultures. These new data are shown in Figure 6a-c.

Reviewer #2

This is an outstanding manuscript.

The authors perform a large scale shRNA screen to identify key regulators of skin stem cell proliferation and SCC formation. They identify YAP and its co-factor WBP2, and proceed to confirm that they interact in the nucleus and function together to drive TEAD-dependent transcription. They further show that the two proteins are co-expressed in stem/progenitor populations in mouse skin and function to promote stem cell proliferation. They then investigate regulation of these factors by Integrin-Src and Rho-kinase signalling in the context of clonal expansion assays and their ability to drive stem cell proliferation.

These findings are novel, as analysis of YAP and WBP2 function in stem cell clonal expansion in skin and SCCs has not been previously performed. The results are therefore an important extension over the existing literature. The quality of the data is very high, with excellent statistical analysis and solidly drawn conclusions.

I have one minor comment that is not essential to address in order for this manuscript to be published.

Figure 5e is a striking result, because it shows that YAP can still be nuclear even when the Src inhibitor Dasatinib is applied, as long as the cell remains flat and basally anchored. This would suggest that a signalling mechanism other than Integrin-Src also operates to sense force/cell shape, which is an important conclusion. The authors further suggest that the effects of Dasatinib might be mediated by stabilisation of AJs that form between groups of cells (Fig S7e), which is an interesting hypothesis (and the reverse of the recently published a-catenin

knockout mouse skin: α E-catenin inhibits a Src-YAP1 oncogenic module that couples tyrosine kinases and the effector of Hippo signaling pathway. Li P, Silvis MR, Honaker Y, Lien WH, Arron ST, Vasioukhin V. *Genes Dev.* 2016 Apr 1;30(7):798-811. doi: 10.1101/gad.274951.115. Epub 2016 Mar 24.)

I am just curious why 2uM Dasatinib was used for these experiments (5e, S7e), rather than the 4uM in the previous experiment (5d), as the effects on YAP localisation in S7e are quite mild. Perhaps this experiment could be repeated with 4 or 5uM Dasatinib for 4 hours, rather than for 24 hours, to test whether YAP localisation might then be affected. It is a little surprising that you can treat NHK cells for 24 hours with these drugs at what is supposedly a high concentration without affecting cell morphology or viability - could the drug become partially degraded by the 24 hour time point?

We used 2 μ M Dasatinib when cells were cultured in KSFM medium, because the cells are more sensitive to the drug than when cultured in serum-containing medium in the presence of feeder cells. This is now stated in the figure legend.

Reviewer #3

In this manuscript, Walko and colleagues report a role of YAP/WBP2 in conferring growth advantage to human epidermal stem cells and in their self-renewal activity. The factors were identified via an extensive genetic screening using normal and neoplastic epidermal cells. While both YAP and WBP2 were shown to be important for clonal expansion, only the former was capable of suppressing onset of terminal differentiation. Mechanistically, contact inhibition acting via ROCK impinged upon Hippo signaling to suppress YAP/WBP2. Therefore, defective contact inhibition leads to unrestrained nuclear YAP/WBP2, leading to uncontrolled growth.

The fact that YAP/WBP conveys pro-growth properties is by itself not profoundly novel because there were papers showing that WBP2 homolog in Fly is a cofactor of Yorkie(YAP homolog) and YAP/TAZ plays a role in promoting SCCs growth.

However, I think that the present data and evidence regarding the role of YAP/WBP in human epithelial stem cell growth is still novel and very informative to be sufficiently published in *Nat Comm*. An unbiased extensive genetic screening approach and immunostaining results extensively revealed the role of YAP/WBP in clonal expansion and differentiation of epidermal cells. In particular, it is important that contact inhibition mediated by stabilization of adherent junction via Rho kinase worked together with Hippo signaling to inhibit YAP/WBP2.

I found that some data seem to be difficult to reconcile with, both as they are and also in light of published findings. To improve the quality of this manuscript, the following are comments I would like to suggest.

1) Having identified YAP/TAZ as key players in conferring growth advantage in epidermal cells, the authors simply assume TEAD dependency and denote future target gene analysis as 'YAP/TAZ/TEAD' or 'YAP/WBP2/TEAD'-dependent. However, the status of TEAD1/2/3/4 themselves have not been shown from the results of the genetic screen. If TEADs were not one of the highest hits comparable to YAP/TAZ/WBP2, the authors need to explain why this may be the case. I guess this may be due to multiple TEADs in cells. A related comment is to complement some key assays (i.e. colony formation, qPCR for differentiation markers) with TEADs knockdown.

We found TEAD4 among the positive growth regulators of NHKs, and TEAD3 and TEAD4 among the positive growth regulators of SCC13 cells in our screens (Supplementary Tables 3 and 9) and have now analysed the functional consequences of knocking-down TEAD1, TEAD3 and TEAD4 (the three TEADs that are expressed in NHKs) in NHKs. We found that knock-down of each TEAD protein reduced clonal growth to a comparable extent. Lastly, we found that TEAD proteins are differentially expressed in cSCC cells compared to NHKs. These new data are shown in the revised Figure 5g-k.

2) The results in Fig.2d-I had been illustrated in other studies (Chan et al, *Oncogene* 2011; Zhang et al, *Cell Death & Differentiation* 2011) that YAP and WBP2 physically interact and the latter enhances the activity of the former. A more interesting question in the reviewer's opinion is whether the association between YAP/TEAD-WBP2 is increased in SCCs compared to normal epidermal cells or under conditions in which clonal growth is preferred (e.g. in Y27632-treated conditions, at least according to this manuscript; conversely, is this interaction decreased in abortive colonies comprising terminally-differentiated cells?).

We now demonstrate by *in situ* PLA that the YAP-WBP2 complex is more abundant in the nucleus of SCC13 cells than NHKs (Figure 2f).

3) Contact inhibition-related experiments presented in this study do not seem to be in line with the current understanding of mechanotransduction operating in the Hippo pathway. Although dasatinib has been shown to be a negative regulator of YAP (Rosenbluh et al, *Cell* 2012), Y27632 has also been shown to be a negative regulator of YAP by promoting YAP cytoplasmic translocation in the skin (Elbediwy et al, *Development* 2016). However, the current manuscript shows that Y27632 rather promotes clonal growth. The authors should therefore describe this discrepancy, and show why such an opposite phenomenon occurs. Perhaps it is because in this particular context YAP/WBP2 are not profoundly affected by canonical Hippo signaling including LATS kinases, but the authors nevertheless need to clarify this point (i.e. most simply, checking whether Y27632 treatment does not reduce YAP/TEAD target genes). A related question is, although the authors show LATS-induced YAP phosphorylation plays a minor role in functionally regulating the localization and activity of YAP, the authors claimed that the subcellular localization of YAP/WBP2 changes in response to adherens junction stabilization (i.e. Fig. 5g). What could be mediating such nucleo-cytoplasmic translocation irrespective of LATS?

We agree that our results are not consistent with the current concept of the role of mechanotransduction in the regulation of YAP because we see effects of Dasatinib/Y27632

on YAP in the context of intercellular adhesion but not in single cells (Figure 7 and Figure S10). Although our results contradict those of (Dupont, S. *et al. Nature* **474**, 179-83 (2011)), they are fully consistent with those of (Das, A., Fischer, R.S., Pan, D. & Waterman, C.M. *The Journal of Biological Chemistry* **291**, 6096-6110 (2016)), now cited in the text.

The study by Elbediwy *et al. (Development* **143**, 1674-1687 (2016)) used HaCaT cells as a cell model of human keratinocytes. HaCaT cells are a clonal immortalized and transformed human keratinocyte cell line and thus are not fully representative of normal human keratinocytes (Boukamp, P. *et al. Journal of Cell Biology* **106**, 761-771 (1988); Sprenger, A. *et al. MCP* **12**, 2509-2521 (2013)). Our experiments investigated longterm (24 hours to 14 days) exposure to low doses (10 μ M) of Y27632, while Elbediwy *et al.* used a high dose (100 μ M) over a short period of time (2 hours) and only observed a moderate effect on YAP localisation.

Several studies have shown that ROCK inhibition promotes clonal expansion of NHKs in culture (e.g. Roshan, A. *et al. Nature cell biology* (2015); Chapman, S., McDermott, D.H., Shen, K., Jang, M.K. & McBride, A.A. *Stem cell research & therapy* **5**, 60 (2014)). We confirm this observation in Figure 7, and show that Dasatinib has the opposite effect. Y27632 treatment did not reduce YAP/TAZ/TEAD target gene expression (Supplementary Fig. 10e), consistent with the lack of an effect on nuclear localization of YAP under these conditions (Supplementary Fig. 10g). We further show that Y27632 treatment, although promoting nuclear re-entry of YAP/WBP2 in the centre of contact-inhibited mature stem cell colonies (Fig. 7d), does not affect the activity of LATS1 (Fig. 7g).

We believe that LATS1/2-independent nucleo-cytoplasmic translocation of YAP is mediated (at least in part) by binding to α -catenin and 14-3-3 proteins (leading to cytoplasmic sequestration of YAP) in response to contact inhibition, consistent with the work of Schlegelmilch *et al. (Cell* **144**, 782-795 (2011)). New data included in Supplementary Fig. 11c show that we can pheno-copy the effect of Y27632 treatment by reducing the calcium concentration in the growth medium to block formation of stable adherens junctions.

4) The authors claim that they found no evidence of canonical Hippo signaling in regulation of clonal epidermal stem cell growth (Fig. 1j, S3g), since knock down of LATS1 or 2 did not increase YAP transcriptional output. However, it is known that LATS 1 and 2 are functionally redundant in some organ for example in the liver. I guess it may also be true in skin. So, it seems hasty to exclude the role of LATS in regulating YAP via depletion of one upstream negative regulator alone.

Schlegelmilch *et al. (Cell* **144**, 782-795 (2011)) showed previously that the canonical Hippo pathway is not involved in regulating YAP in confluent keratinocytes. We confirm these observations in Supplementary Fig. 4e-g and Supplementary Fig. 9 and show that combined shRNA-mediated knockdown of LATS1 and 2 does not affect expression of YAP target genes or sub-cellular distribution of YAP. Our findings are in agreement with those of Aragona *et al. (Cell* **154**, 1047-1059 (2013)), who showed that upon contact inhibition mechanical cues imposed by cell-cell adhesion and cellular crowding dominate over canonical Hippo signalling in regulating YAP.

Besides, in Fig. S3f, the level of LATS1 protein seems to correlate with that of p-YAP (S127) (and inversely correlated with total YAP), implying a regulatory role of LATS in YAP activity in this context. So, unless authors knock down both LATS1/2 in this setting, authors can not rule out a possibility that canonical Hippo signaling including LATS1/2 affect YAP functions in cell growth.

We have now shown that combined knock down of LATS1 and 2 does not affect NHK proliferation and terminal differentiation during epidermal reconstitution (Fig. 6a-c).

It is note that canonical Hippo signaling including LATS1/2 may affect YAP function in cell differentiation but not in growth. In fact, mice lacking Sav1 (WW45) showed that mutant skin had a an expansion of Stem/progenitor cells with less differentiation (Lee et al, EMBO J. 2008 27(8):1231-42. This paper showed that Hippo pathway clearly is activated to inhibit YAP function during epithelial differentiation. Based on these, authors need to include this possible role of canonical Hippo signaling in regulation of epidermal stem cell growth or differentiation.

We have now cited this paper. However, our own experiments show that while YAP plays a role in regulating both proliferation and terminal differentiation of NHKs, LATS1 and 2 do not. We believe that, as stated above, stablisation of adherens junctions is the major regulator of YAP in the context of contact inhibited NHKs.

REVIEWERS' COMMENTS:

Reviewer #1 (Remarks to the Author):

I have had a chance to look at the manuscript again and I recommend publication. The authors addressed all of my concerns and included extensive new data.

Reviewer #2 (Remarks to the Author):

I am satisfied with the revisions. The authors have responded well to all three reviewers and the manuscript is now suitable for publication in Nature Communications.

Reviewer #3 (Remarks to the Author):

I found that my concerns have been successfully addressed by the authors. This paper is very important and novel enough to be published.

Response to referees

REVIEWERS' COMMENTS:

Reviewer #1 (Remarks to the Author):

I have had a chance to look at the manuscript again and I recommend publication. The authors addressed all of my concerns and included extensive new data.

Reviewer #2 (Remarks to the Author):

I am satisfied with the revisions. The authors have responded well to all three reviewers and the manuscript is now suitable for publication in Nature Communications.

Reviewer #3 (Remarks to the Author):

I found that my concerns have been successfully addressed by the authors. This paper is very important and novel enough to be published.

There were no issues raised by the referees.